# Manifold Regularization for Locally Stable Deep Neural Networks

## Abstract

We apply concepts from manifold regularization to develop new regularization techniques for training locally stable deep neural networks. Our regularizers encourage functions which are smooth not only in their predictions but also their decision boundaries. Empirically, our networks exhibit stability in a diverse set of perturbation models, including $\ell_2$, $\ell_\infty$, and Wasserstein-based perturbations; in particular, against a state-of-the-art PGD adversary, a single model achieves both $\ell_\infty$ robustness of 40% at $\epsilon = 8/255$ and $\ell_2$ robustness of 48% at $\epsilon = 1.0$ on CIFAR-10. We also obtain state-of-the-art verified accuracy of 21% in the same $\ell_\infty$ setting. Furthermore, our techniques are efficient, incurring overhead on par with two additional parallel forward passes through the network; in the case of CIFAR-10, we achieve our results after training for only 3 hours, compared to more than 70 hours for standard adversarial training.

## 1 Introduction

Recent results in deep learning highlight the remarkable performance deep neural networks can achieve on tasks using data from the natural world, such as images, video, and audio. Though such data inhabits an input space of high dimensionality, the physical processes which generate the data often manifest significant biases, causing realistic inputs to be sparse in the input space.

One way of capturing this intuition is the manifold assumption, which states that input data is not drawn uniformly from the input space, but rather supported on some smooth submanifold(s) of much lower dimension. Starting with the work of Belkin et al. (2006), this formulation has been studied extensively in the setting of semi-supervised kernel and regression methods, where algorithms exploit the unlabelled data points to learn functions which are smooth on the input manifold (Geng et al., 2012; Goldberg et al., 2008; Niyogi, 2013; Sindhwani et al., 2005; Tsang and Kwok, 2007; Xu et al., 2010). Such techniques have seen less use in the context of deep neural networks, owing in part to the ability of such models to generalize from relatively sparse data (Zhang et al., 2016).

**Contributions**  We apply concepts from manifold regularization to train locally stable deep neural networks. In light of recent results showing that neural networks suffer widely from adversarial inputs (Szegedy et al., 2013), our goal is to learn a function which does not vary much in the neighborhoods of natural inputs, independently of whether the network classifies correctly. We show that this definition of local stability has a natural interpretation in the context of manifold regularization, and propose an efficient regularizer based on an approximation of the graph Laplacian when the data is sparse, i.e., the pairwise distances are large. Crucially, our regularizer exploits the continuous piecewise linear nature of ReLU networks to learn a function which is smooth over the data manifold in not only its outputs but also its decision boundaries.

We evaluate our approach by training neural networks with our regularizers for the task of image classification on CIFAR-10 (Krizhevsky et al., 2009). Empirically, our networks exhibit robustness against a variety of adversarial models implementing $\ell_2$, $\ell_\infty$, and Wasserstein-based attacks. We also achieve state-of-the-art *verified* robust accuracy under $\ell_\infty$ of size $\epsilon = 8/255$. Furthermore, our regularizers are cheap: we simply evaluate the network at two additional random points for each training sample, so the total computational cost is on par with three parallel forward passes through the network. Our techniques thus present a novel, regularization-only approach to learning robust

neural networks, which achieves performance comparable to existing defenses while also being an order of magnitude more efficient.

## 2 BACKGROUND

**Manifold regularization**  The manifold assumption states that input data is not drawn uniformly from the input domain $\mathcal{X}$, also know as the *ambient space*, but rather is supported on a submanifold $\mathcal{M} \subset \mathcal{X}$, called the *intrinsic space*. There is thus a distinction between regularizing on the ambient space, where the learned function is smooth with respect to the entire input domain (e.g., Tikhonov regularization (Phillips, 1962; Tikhonov et al., 2013)), and regularizing over the intrinsic space, which uses the geometry of the input submanifold to determine the regularization norm.

A common form of manifold regularization assumes the gradient of the learned function $\nabla_{\mathcal{M}} f(x)$ should be small where the probability of drawing a sample is large; we call such functions "smooth". Let $\mu$ be a probability measure with support $\mathcal{M}$. This leads to the following intrinsic regularizer:

$$||f||_I^2 := \int_{\mathcal{M}} ||\nabla_{\mathcal{M}} f(x)||^2 d\mu(x) \tag{1}$$

In general, we cannot compute this integral because $\mathcal{M}$ is not known, so Belkin et al. (2006) propose the following discrete approximation that converges to the integral as the number of samples grows:

$$||f||_I^2 \approx \frac{1}{N^2} \sum_{i,j=1}^{N} (f(x_i) - f(x_j))^2 L_{i,j} \tag{2}$$

Here, the $x_1, ..., x_N$ are samples drawn, by assumption, from the input manifold $\mathcal{M}$ according to $\mu$, and $L$ is a matrix of weights measuring the similarity between samples. The idea is to approximate the continuous input manifold using a discrete graph, where the vertices are samples, the edge weights are distances between points, and the Laplacian matrix $L$ encodes the structure of this graph. A common choice of weights is a heat kernel: $L_{i,j} = L(x_i, x_j) := \exp(-||x_i - x_j||^2/s)$. To improve computational costs, weights are often truncated to the $k$-nearest neighbors or within some $\epsilon$-ball. Note that the Laplacian can also be interpreted as a discrete matrix operator, which converges under certain conditions to the continuous Laplace operator (Belkin and Niyogi, 2008).

**ReLU networks**  Our development focuses on a standard architecture for deep neural networks: fully-connected feedforward networks with ReLU activations. In general, we can write the function represented by such a network with $n$ layers and parameters $\theta = \{A_i, b_i\}_{i=1,...,n-1}$ as

$$z_0 = x \tag{3}$$
$$\hat{z}_i = A_i \cdot z_{i-1} + b_i \qquad \text{for } i = 1, ..., n-1 \tag{4}$$
$$z_i = \sigma(\hat{z}_i) \qquad \text{for } i = 1, ..., n-2 \tag{5}$$
$$f(x; \theta) = \hat{z}_{n-1} \tag{6}$$

where the $A_i$ are the weight matrices and the $b_i$ are the bias vectors. We call the $z_i$ "hidden activations", or more simply, activations, and the $\hat{z}_i$ "pre-activations".

In this work, we consider networks in which $\sigma(\cdot)$ in (5) is the Rectified Linear Unit (ReLU)

$$z_i = \sigma(\hat{z}_i) := \max(0, \hat{z}_i) \tag{7}$$

It is clear from this description that ReLU networks are a family of continuous piecewise linear functions. We denote the linear function induced by an input $x$ as $f_x(\cdot; \theta)$, i.e., the analytic extension of the local linear component about $x$ over the input domain.

**Adversarial robustness**  One common measure of robustness for neural networks is against a norm-bounded adversary. In this model, the adversary is given an input budget $\epsilon$ over a norm $||\cdot||_{in}$, and asked to produce an output perturbation $\delta$ over a norm $|\cdot|_{out}$. A point $x'$ is an $\epsilon$-$\delta$ adversarial example for an input pair $(x, y)$ if

$$||x' - x||_{in} \leq \epsilon \tag{8}$$
$$|f(x'; \theta) - y|_{out} \geq \delta \tag{9}$$

When the specific norm is either unimportant or clear from context, we also write the first condition as $x' \in N_\epsilon(x)$, where $N_\epsilon(x)$ refers to the $\epsilon$-ball or neighborhood about $x$. If such an adversarial example does not exist, we say that the network is $\epsilon$-$\delta$ robust at $x$. Standard examples of $|| \cdot ||_{in}$ include the $\ell_2$ and $\ell_\infty$ "norm", defined for vectors as $||x||_\infty := \max_i |x_i|$. For classification tasks, the adversary is successful if it produces an example in the $\epsilon$-neighborhood of $x$ which causes the network to misclassify. In this case, we drop $\delta$ and say that the network is $\epsilon$-robust at $x$. Note that if $f(x; \theta)$ is already incorrect, then $x$ suffices as an adversarial example.

## 3 RELATED WORK

Manifold regularization was first introduced by Belkin et al. (2006) in the context of semi-supervised learning, where the goal was to leverage unlabeled samples to learn a function which behaves well (e.g., is smooth, or has low complexity) over the data manifold. The use of manifold regularization for deep neural networks has been explored in several contexts (Tomar and Rose, 2014; 2016; Hu et al., 2018; Zhu et al., 2018). In particular, Lee et al. (2015) combine manifold regularization with adversarial training and show improvements in standard test accuracy. Our approach is to use manifold regularization to induce stability separately from accuracy. We note that a similar decomposition between accuracy and stability forms the basis for the TRADES algorithm (Zhang et al., 2019), though the training procedure ultimately relies on adversarial training. Hein and Andriushchenko (2017) propose a conceptually similar regularizer to minimize the difference between logits and show improved $\ell_2$ certified robustness. Finally, several prior works explore regularizing for robustness based on other differential forms (Ross and Doshi-Velez, 2017; Jakubovitz and Giryes, 2018), though they only report results using the weaker single-step PGD adversary. In particular, a recent work by Zhai et al. (2020) uses randomized smoothing for $\ell_2$ certified robustness, and claim similar computational advantage due to avoiding adversarial training, but still take 61 hours to train, compared to only 3 hours in our approach.

Adversarial examples were introduced by Szegedy et al. (2013), who found that naively trained neural networks suffer almost complete degradation of performance on natural images under slight perturbations which are imperceptible to humans. A standard class of defenses is *adversarial training*, which is characterized by training on adversarially generated input points (Goodfellow et al., 2014). In particular, the Projected Gradient Descent (PGD) attack (Kurakin et al., 2016; Madry et al., 2017) is widely considered to be an empirically sound algorithm for both training and evaluation of robust models. However, such training methods rely on solving an inner optimization via an iterative method, effectively increasing the number of epochs by a multiplicative factor (e.g., an overhead of 5–10x for standard PGD). Achieving robustness against multiple adversarial models has also been explored previously (Schott et al., 2018; Tramer and Boneh, 2019; Maini et al., 2019; Croce and Hein, 2019b), though in most cases these works use weaker variants of the subset of standard adversaries we consider (e.g., a smaller $\epsilon$ or the single-step version of PGD).

Another approach is to train models which are *provably robust*. One method is to use an exact verification method, such as an MILP solver, to prove that the network is robust on given inputs (Tjeng et al., 2017). In particular, Xiao et al. (2019) use a similar loss based on ReLU pre-activations to learn stable ReLUs for efficient verification, but rely on a PGD adversary to train a robust model. Certification methods modify models to work directly with neighborhoods instead of points (Dvijotham et al., 2018; Gowal et al., 2018; Mirman et al., 2018; Wong et al., 2018). In practice, the inference algorithms must overapproximate the neighborhoods to preserve soundness while keeping the representation compact as it passes through the network. This strategy can be interpreted as solving a convex relaxation of the exact verification problem. Though certification thus far has produced better lower bounds, verification as a technique is fully general and can be applied to any model (given sufficient time); recent work also suggests that methods using layerwise convex relaxations may face an inherent barrier to tight verification (Salman et al., 2019).

## 4 SETTING

We reframe the goal of learning functions that are robust using a perspective which decouples stability from accuracy. The key observation is that we would like to train networks that are locally stable around natural inputs, even if the network output is incorrect. This approach contrasts with

adversarial training, which attempts to train the network to classify correctly on worst-case adversarial inputs. In particular, recall that a network is $\epsilon$-robust at $x$ if no point in the $\epsilon$-neighborhood of $x$ causes the network to misclassify. We consider the related property of $\epsilon$-stability (cf. Zheng et al. (2016)):

**Definition 4.1.** *A function $f$ is $\epsilon$-$\delta$ stable at an input $x$ if for all $x' \in N_\epsilon(x)$, $|f(x) - f'(x)| \leq \delta$. A classifier $f$ is $\epsilon$-stable at an input $x$ if for all $x' \in N_\epsilon(x)$, $f(x) = f(x')$.*

As $\epsilon$-stability is independent of the correct label for $x$, we argue that $\epsilon$-stability is a property of the function with respect to the input manifold and can thus be captured using manifold regularization. For completeness, we state the following connection between robustness and stability:

**Proposition 4.1.** *A function $f$ is $\epsilon$-$\delta$ robust at an input $x$ iff $f$ is $\epsilon$-$\delta$ stable at $x$ and $f(x) = y$. A classifier $f$ is $\epsilon$-robust at an input $x$ iff $f$ is $\epsilon$-stable at $x$ and $f$ correctly classifies $x$.*

## 5 MANIFOLD REGULARIZATION FOR DEEP NEURAL NETWORKS

Applying the regularization term in (2) yields, in the limit, a function which is smooth on the data manifold. Unfortunately, a straightforward approach does not suffice for our goal of learning $\epsilon$-stable deep neural networks. The first problem is that smoothness on the data manifold does not yield $\epsilon$-stability (Stutz et al. (2019)); indeed, we are concerned with the behavior of our function in an $\epsilon$-neighborhood of the manifold, not just on the manifold. The second observation is that convergence of the discrete approximation requires that the samples be dense over the input manifold; however, this assumption is almost certainly violated in most practical applications, particularly in the deep learning regime. The next two sections are dedicated to these challenges.

### 5.1 RESAMPLING FOR LOCAL SMOOTHNESS

We write the $\epsilon$-neighborhood of a manifold $\mathcal{M}$ as $\mathcal{M}_\epsilon := \{x : \exists y \in \mathcal{M}, ||x - y|| \leq \epsilon\}$. Since $\epsilon$-stability is defined over the $\epsilon$-neighborhood of every input point, we might want a function which is smooth over $\mathcal{M}_\epsilon$ instead of just $\mathcal{M}$, e.g., by slightly perturbing every input point. This strategy does produce samples from the $\epsilon$-neighborhood of the data manifold, however note that the induced distribution is not uniform but rather the convolution of the choice of noise distribution with the uniform distribution over $\mathcal{M}$. Nevertheless, this procedure exhibits several properties we can exploit. The first is that for sufficiently small $\epsilon$, we get nearly the same operator over $\mathcal{M}_\epsilon$ and $\mathcal{M}$, so that smoothness over $\mathcal{M}_\epsilon$ does not sacrifice smoothness on the original data manifold $\mathcal{M}$. A more subtle property is that we can actually draw as many distinct points as we would like from $\mathcal{M}_\epsilon$. We leverage these extra points to build a regularizer which yields good estimations of the local smoothness. Moreover, taking a discrete approximation of the form in Equation 2 with the resampled points from $\mathcal{M}_\epsilon$ still converges to the original operator. Formally, we can state the following:

**Proposition 5.1.** *Let $\epsilon, s > 0$ be given. Let $x_1, ..., x_n$ be $n$ samples drawn uniformly at random from a submanifold $\mathcal{M} \subset \mathcal{X}$. For each $x_i$, pick $c$ new points $x_{i,1}, ..., x_{i,c}$ by sampling iid perturbations $\delta_{i,1}, ..., \delta_{i,c}$ and setting $x_{i,j} = x_i + \delta_{i,j}$, where $\forall i, j, ||\delta_{i,j}|| < \epsilon$. Given a kernel $k_s(x, y) = \exp(-||x - y||^2/s)$, let $L$ and $L_c$ be the Laplacian matrices defined by the $n$ original samples and $n \cdot c$ new samples, respectively. Then if $\epsilon^2 < s$, we have that $L$ and $L_c$ converge to the same operator in the limit as $n \to \infty$.*

We prove this result in Appendix A. For our purposes, this result states that the resampled Laplacian enjoys the same behavior in the limit as the original Laplacian.

### 5.2 SPARSE APPROXIMATION OF RESAMPLED LAPLACIAN

The Laplacian matrix is dense, and requires both $O(N^2)$ time and space to compute and store. It is thus standard to employ heuristics for sparsifying Laplacians to derive efficient algorithms; in this work, we consider the $\epsilon$-neighborhood sparsifier, denoted $L_\epsilon$, which is defined as the Laplacian of the graph created by retaining only those edges whose weights are at most $\epsilon$ (Von Luxburg, 2007).

To motivate this approach, our main observation is that, under certain sparsity assumptions in the data, the resampled Laplacian will consist of two types of edges: very short edges between points

resampled from the same $\epsilon$-neighborhood, and very long edges between points sampled from different $\epsilon$-neighborhoods. For an appropriate choice of the scale factor $s$, the exponential form of the heat kernel causes the weights on the long edges to fall off very quickly compared to the short edges. The following results gives bounds for this approximation of the regularizer in Equation 2 under a certain separation assumption in the data:

**Proposition 5.2.** *Consider a function $f$ and a set of points $\{x_i\}_{i=1}^N$ such that the pairwise differences bounded from above and below by $0 < c_1 \leq ||f(x_i) - f(x_j)||_2^2 \leq c_2 < \infty$ for all $i \neq j$. Let $s$ be the parameter of the heat kernel $k_s = \exp(-||x_i - x_j||^2/s)$. Assume further that the points $\{x_i\}$ are separated in the sense that there exist at least $O(n)$ pairs of points $(x_{i1}, x_{i2}), i \in S$ whose distances are bounded from above by $\sqrt{s}$, and the remainder of the points have distances bounded from below by $m\sqrt{s}$ for some constant $m > 1$. Writing $R(L) := \sum_{i,j=1}^N (f(x_i) - f(x_j))^2 L_{i,j}$, we have that*

$$R(L) - c_2 N^2 \exp(-m^2) \leq R(L_\epsilon) \leq R(L) \leq (1 + b^{-1})R(L_\epsilon) \tag{10}$$

*where $b = \Theta((c_1/c_2)(\exp(m^2)/N))$.*

We give a proof in Appendix A. In this work, we only consider bounded functions $f$, which allows us to apply this proposition. This result gives two bounds on the dense regularization $R(L)$ in terms of $R(L_\epsilon)$. The first inequality says that the *absolute* approximation error is bounded by a term which vanishes unless the *squared* sample count $N^2$ grows as the exponential of the squared separation $m^2$. The third inequality gives a tighter bound on the *relative* error in a term that vanishes unless the *linear* sample count $N$ grows as the exponential of the squared separation $m^2$, but requires that we have good control over the ratio $c_1/c_2$. Note that the dependence on $c_1/c_2$ is unavoidable in the sense that a function which is very smooth in local neighborhoods of size $s$ will necessarily have large relative contribution from the longer edges; however, in this case we still have a bound on the absolute contribution (though we trade the ratio $c_1/c_2$ for a factor of $N$). Crucially, in both cases the error bound depends on an exponential factor of the squared separation $m^2$.

To provide some empirical evidence that the separation $m$ can be made non-trivial in some settings, we offer the following statistics from CIFAR-10. We select a random subset of 10,000 (20%) training samples. For each of these points, we find the closest point within the remaining 49,999 training samples, and compute the distance. Using the $\ell_2$ metric, we found a mean distance of 9.597, a median distance of 9.405, and the distance at the 10th percentile to be 5.867. Conversely, we resample points in an $\ell_\infty$ ball of radius $\epsilon = 8/255$, which is contained in an $\ell_2$ ball of radius $\sqrt{3 \cdot 32 \cdot 32} \cdot 8/255 = 1.739$. In fact, only 21 training samples, or 0.21% of the random subset, have a partner closer than twice the perturbation bounds. Thus, after the resampling procedure, setting the scale parameter to $s = \epsilon^2$ and taking the $\epsilon$-neighborhood Laplacian should yield a good approximation to the full Laplacian. Finally, to generalize this to larger datasets, for fixed $s$ one would expect that the separation $m\sqrt{s}$ between points grows with the dimension of the ambient space, which gives an exponential decay in the approximation error. Indeed, this is one of the manifestations of the oft-cited curse of dimensionality (Hastie et al., 2009).

Due to our resampling procedure, we have at least $c^2 N$ points whose squared pairwise distances are less than $\epsilon^2$, which motivates setting $s \approx \epsilon^2$; notice that this choice of $s$ satisfies the assumptions of Proposition 5.1 needed for convergence. However, rather than find the $\epsilon$-neighborhood Laplacian, we propose to compute the regularizer only using points resampled from the same $\epsilon$-neighborhood, yielding the following sparse approximation of the intrinsic regularizer:

$$||f||_I^2 \approx \frac{1}{c^2 N} \sum_{i=1}^N \sum_{j,k=1}^c (f(x_{i,j}) - f(x_{i,k}))^2 L(x_{i,j}, x_{i,k}) \tag{11}$$

We emphasize that this is a heuristic, motivated by computational efficiency, whose quality ultimately depends on the sparsity of the dataset and the particular choice of $\epsilon$. In particular, this approximation should not be used when the data is dense and there are many points that are within $\epsilon$ distance of each other. However, the diagonal form of this approximation permits extremely efficient computations, particularly on vectorized hardware.

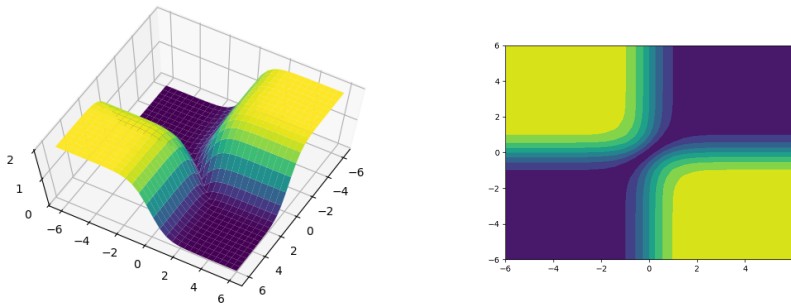

Figure 1: Surface and contour plots for $H_\alpha$ ($\alpha = 1$).

## 5.3 HAMMING REGULARIZERS

We additionally leverage the structure of ReLU networks to induce a stronger regularization effect on the data manifold. The central observation is that not just do we want the function computed by the neural network to be constant (i.e., smooth with respect to its outputs) on the manifold, we also want the network to produce its outputs in roughly the same way.

First, we identify every local linear component $f_x(\cdot; \theta)$ with its "activation pattern", i.e., the sequence of branches taken at each ReLU. We will write $f_H(x; \theta)$ for the map that takes inputs $x$ to their activation patterns, which live on the unit hypercube $\{0, 1\}^N$ (where $N$ is the number of ReLUs in the network). If we endow the hypercube with the standard Hamming distance $d_H$, this induces a pullback (psuedo-)metric in the input space: $d_H^*(x, y) = d_H(f_H(x; \theta), f_H(y; \theta))$. Notice that the metric identification is exactly the set of local linear components $f_x(\cdot; \theta)$.

Recall that the goal of regularization is to reduce the complexity of the learned function $f$ in the $\epsilon$-neighborhood of inputs $x$. We argue that $d_H^*$ provides a concise measure of this complexity. Specifically, if we consider the number of distinct local linear components in $N_\epsilon(x)$, then since $d_H^*$ is a psuedo-metric, we have that $\forall x' \in N_\epsilon(x)$, $d_H^*(x, x'; \theta) = 0$ if and only if $N_\epsilon(x)$ is a single linear component. Thus, minimizing $d_H^*$ between $x$ and all $x' \in N_\epsilon(x)$ reduces the number of linear components in the neighborhood of $x$. In fact, by the triangle inequality, the same holds in the interior of any convex polytope defined by a set of points $\{x_i\}_{i \in I}$ where $\forall i, i' \in I, d_H^*(x_i, x_{i'}; \theta) = 0$. This makes minimizing $d_H^*$ very sample efficient.

Treating $f_H(\cdot; \theta)$ as an output of the network, we have the following (sparse) manifold regularizer:

$$||f_H(\cdot; \theta)||_I^2 := \int_{\mathcal{M}_\epsilon} ||\nabla_{\mathcal{M}_\epsilon} f_H(x; \theta)||^2 d\mu(x) \tag{12}$$

$$\approx \frac{1}{c^2 N} \sum_{i=1}^N \sum_{j,k=1}^c d_H^*(x_{i,j}, x_{i,k}; \theta)^2 L(x_{i,j}, x_{i,k}) \tag{13}$$

which is just Equations 1 and 11 with the outputs $f(x)$ replaced by the activation pattern map. However, this loss term is not continuous in the inputs, and furthermore, the gradients vanish almost everywhere, so it does not generate good training signals. We thus use a continuous relaxation:

$$H_\alpha(\hat{z}, \hat{y}; \theta) := \mathrm{abs}(\tanh(\alpha * \hat{z}) - \tanh(\alpha * \hat{y})) \tag{14}$$

This form is differentiable everywhere except when $\hat{z} = \hat{y}$, and recovers the Hamming distance when we take $\alpha$ to infinity (after scaling). Qualitatively, sensitive activations (i.e., small $|\hat{z}|$ and $|\hat{y}|$) are permitted so long as they are precise. Figure 1 presents the surface and contour plots of $H_\alpha$. Note that this idea can be extended more generally to other activation functions by penalizing differing pre-activations more when the second derivative of the activation function is large (and so the first-order Taylor approximation has larger errors).

## 5.4 TRAINING WITH SPARSE MANIFOLD REGULARIZATION

For every sample $x$, we generate a new random maximal perturbation $\rho \in \{\pm\epsilon\}^d$ and produce the pair of perturbed inputs $x^+ := x + p$ and $x^- := x - p$. We compute the standard manifold regularization term as $||f(\cdot, \theta)||_I^2 \propto \sum_i ||f(x_i^+) - f(x_i^-)||_2^2$. For the Hamming regularizer, we use the $\ell_2$ norm to combine the Hamming distances between pre-activations within a layer and normalize by twice the number of elements in each layer. We sum over the layers, then normalize by the total number of layers. Note that in both cases, the weights $L(x_i^+, x_i^-)$ can be dropped since the distance between $x_i^+$ and $x_i^-$ is constant. The final optimization objective is thus

$$\theta^* = \arg\min_\theta \frac{1}{N} \sum_{i=1}^N V(f(x_i; \theta), y_i) + \gamma_K ||f(\cdot; \theta)||_K^2 + \gamma_I ||f(\cdot; \theta)||_I^2 + \gamma_H ||f_H(\cdot; \theta)||_I^2 \quad (15)$$

where $V$ is the loss function, $||f(\cdot, \theta)||_K^2$ is the ambient regularizer (e.g., $\ell_2$ regularization), and the $\gamma_K, \gamma_I, \gamma_H$ are hyperparameters which control the relative contributions of the different regularizers.

## 5.5 DISCUSSION

Our development has focused on learning deep neural networks which are smooth when samples are sparse over the input manifold $\mathcal{M}$. While this property is related to $\epsilon$-stability, in general the two are not equivalent. Roughly, $\epsilon$-stability is inherently a property about worst case behavior, whereas manifold regularization is aimed at learning a function which is smooth on average. Nonetheless, we note that a function achieving zero loss with the manifold regularizer yields perfect stability (i.e., a constant function on the data manifold).

Next, we discuss our choice to focus on the sparse setting. The basic idea is two-fold: first, the requirement that the learned function be $\epsilon$-stable significantly alters the geometry of the input manifold; second, when data is sparse, the amount of information one can glean about the overall geometry of the input manifold is limited, as is evidenced by the vanishing weights in the Laplacian. Our main hypothesis is that the combination of these two observations allows one to formulate a version of the regularizer built from resampled data, which maintains the same properties in the limit but yields more local information when data is sparse.

Conversely, consider the alternative approach, namely, directly learning a smooth function when it is possible to produce samples, not necessarily labelled, that are dense over the input manifold $\mathcal{M}$. Then given the central thesis behind this work, one would expect stability to improve. In fact, the top four results for robust accuracy reported in Croce and Hein (2020) all exploit an additional dataset of unlabeled images which is orders of magnitude larger than the original training set (Carmon et al., 2019; Hendrycks et al., 2019; Uesato et al., 2019; Wang et al., 2020).

## 6 EXPERIMENTAL RESULTS

We report results for two models trained on CIFAR-10 image classification using our regularization techniques: a smaller CNN for exact verification, following Xiao et al. (2019); and a PreActRes-Net18 model (He et al., 2016) for benchmarking against a variety of adversarial models. We also include three ablation studies for the larger model, including one which does not use the sparsification scheme developed in Section 5.2 (dense regularizer), and two using the individual regularizers to isolate their individual effects on stability (intrinsic / Hamming regularizer only). Appendix B reports training details and hyperparameters; full experimental results (including MNIST, which is not discussed here) are in Appendix C.

## 6.1 STABILITY

The main characteristic that differentiates our approach conceptually from existing methods for adversarial robustness is that we train our models using pure regularization, i.e., without reference to a particular adversary. In comparison, most methods train and test using inputs produced by the same adversary. While this procedure yields good performance against the adversary used during training, performance degrades significantly when evaluated under different perturbation models. In general, the bounds for various adversarial models are set so that maximal perturbations remain

Table 1: CIFAR-10 robust accuracy against various adversaries and perturbation models.

| Defense | Test Accuracy (%) | | | |
| --- | --- | --- | --- | --- |
| | $\ell_2$ $\epsilon = 0.5$ | $\ell_\infty$ $\epsilon = 8/255$ | Wasserstein $\epsilon = .1$ | Clean |
| Wong et al. (2019) | | | 76 | 80.69 |
| baseline ($\ell_\infty$ certified) | | | 61 | 66.33 |
| Maini et al. (2019) | 66.0 | 49.8[‡] | | 81.7 |
| baseline ($\ell_\infty$ PGD, Madry et al. (2017)) | <5.0 | 45.8 | | 87.3 |
| **Manifold Regularization** | 57.53[§] | 36.90[§] | 66.02 | 69.95 |

[§]Computed using the full AutoAttack+ attack (Croce and Hein, 2020), which includes APGD-CE, APGD-DLR (+targeted), FAB (+targeted) (Croce and Hein, 2019a), and Square Attack (Andriushchenko et al., 2019). [‡]With slightly smaller $\epsilon = 0.03 \approx 7.65/255$. [†]Reported in Maini et al. (2019).

somewhat reasonable by human standards; thus, we expect a model that is stable in the general sense to be robust against a variety of adversaries, particularly those not seen during training.

In line with this observation, we test a single model trained to be stable using manifold regularization against three norm-bounded perturbation models. The first two constitute the most common settings for adversarial robustness, namely, CIFAR-10 robustness using $\ell_\infty$ and $\ell_2$ PGD adversaries bounded at $\epsilon = 8/255$ and 0.5, respectively. We also use a Wasserstein-based PGD adversary at $\epsilon = 0.1$, which is a metric for images more closely aligned with human perceptions. Our results indicates that a model trained using our manifold regularizer does in fact learn to be more stable on test images across a variety of different perturbation models (Table 1). We obtained our results after training for 3 hours on one GPU, compared to several days for standard PGD training (Shafahi et al., 2019).

For comparison, we note that standard adversarial training using PGD from Madry et al. (2017) , which forms the basis of every state-of-the-art approach for $\ell_\infty$ robustness (Croce and Hein, 2020), achieves negligible performance when evaluated in the $\ell_2$ setting. We also report results from Maini et al. (2019), which was the only other approach in the literature to evaluate their models for both $\ell_\infty$ and $\ell_2$ robustness at the full levels of $\epsilon$ using a PGD adversary; their models are trained for 50 epochs with 50 steps of PGD against multiple adversaries in parallel ($\approx 5000$ effective epochs). Finally, we also report results from Wong et al. (2019) for comparison against the Wasserstein adversary. Their results are obtained against a PGD adversary using a 50-step inner optimization (number of training epochs is not reported); their baseline model is trained to be certifiably robust against $\ell_\infty$ attacks of size $\epsilon = 2/255$, and achieves lower robustness than ours.

## 6.2 ABLATION STUDIES

We run several ablation studies to better understand the behavior of our methods. The first column of Table 2 reports the *verifiable robustness* of our smaller model, which outperforms the previous state of the art as reported in Xiao et al. (2019). We emphasize that we provide this metric as a baseline for establishing a provable lower bound on the robustness achieved by our defense, compared to empirical robustness which is often a brittle measure of true robustness, particularly for newly proposed defenses (Athalye et al., 2018; Carlini et al., 2019; Tramer et al., 2020).

The second column of Table 2 presents robust accuracy against the standard PGD adversary used in the literature. The dense regularizer has both lower robust and clean accuracy, which suggests an over-regularization effect stemming from longer edges that do not yield reliable information about the intrinsic geometry of the manifold. Furthermore, training with the dense regularizer takes around 21.5 hours, or nearly 8 times longer than the sparse method (due to both the overhead of computing additional gradients during backpropagation, as well as the smaller batch sizes for offsetting the increased memory requirements for storing the dense Laplacian). We also report the results from using intrinsic or Hamming regularizers alone, which indicate that both terms contribute jointly and individually to our performance. For comparison, we report results for other approaches which are not variants of PGD (i.e., do not rely on solving a minimax problem at each training loop).

Table 2: CIFAR-10 provable and robust accuracies against an $\ell_\infty$ adversary at $\epsilon = 8/255$.

| Mechanism | Defense | Test Accuracy (%) | | |
|---|---|---|---|---|
| | | Verified | Robust | Clean |
| Adversarial Training | PGD (Madry et al. (2017)) | | **45.8** | 87.3 |
| | FGSM (Madry et al. (2017)) | | 0.0 | 90.3 |
| Verification | Xiao et al. (2019) | 20.27 | 26.78 | 40.45 |
| | **Manifold Regularization (small)** | 21.04 | 25.56 | 36.66 |
| Regularization | Pang et al. (2019)‡ | | 24.8 | **92.7** |
| | **Manifold Regularization (large)** | | 40.54 | 69.95 |
| | dense regularizer | | 35.04 | 64.08 |
| | intrinsic regularizer only | | 20.11 | 34.74 |
| | Hamming regularizer only | | 24.87 | 90.24 |

†With slightly smaller $\epsilon = 0.03 \approx 7.65/255$. §Computed from 500 / 10000 images in test set. ‡Results for regularization only; reported robust accuracy is 55.0% when trained with PGD.

To the best of our knowledge, Pang et al. (2019) is the only other result in the literature which achieves non-trivial $\ell_\infty$ robust accuracy on the standard benchmark of CIFAR-10 at $\epsilon = 8/255$ using a regularization-only approach; their results are better than either of our regularizers used independently, but significantly worse than when the regularizers are used together, in either the sparse or the dense setting. For context, the best result in this setting is reported by Wang et al. (2020) using a PGD variant at 65.04% robust accuracy.

# 7 CONCLUSION

We design regularizers based on manifold regularization that encourage piecewise linear neural networks to learn locally stable functions. We demonstrate this stability by showing that a single model trained using our regularizers is resilient against $\ell_2$, $\ell_\infty$, and Wasserstein-based attacks. We also achieve state-of-the-art verified robustness of 21% against $\ell_\infty$-bounded perturbations of size $\epsilon = 8/255$ on CIFAR-10. Critically, computing our regularizers relies only on random sampling, and thus does not require running an inner optimization loop to find strong perturbations at training time. As such, our techniques exhibit strong scaling, since they increase batch sizes rather than epochs during training, allowing us to train an order of magnitude faster than standard adversarial training. This work thus presents the first regularization-only approach to achieve comparable results to standard adversarial training against a variety of perturbation models.

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

# A  PROOFS

*Proof of Proposition 5.1.* Given a heat kernel $k_s(x, y) = \exp(-||x - y||^2/s)$ and samples $x_1, ..., x_N$, we define the discrete operator $L$ for an out-of-sample point $x$ as

$$Lf(x) = \frac{1}{N} \sum_{i=1}^{N} k_s(x, x_i) f(x) - \frac{1}{N} \sum_{i=1}^{N} k_s(x, x_i) f(x_i) \tag{16}$$

Our claim is that the original Laplacian $L$ and the resampled Laplacian $L'_c$ converge pointwise to the same continuous operator.

The proof proceeds in two parts. First, we show that resampling once still yields convergence of the Laplacian. Notice that we can model the resampling procedure as a form of noise; our desired result follows immediately from the following result of Coifman and Lafon (2006):

**Proposition A.1** (Criterion 5 in Coifman and Lafon (2006)). *Suppose that $\mathcal{X}$ is a perturbed version of $\mathcal{M}$, that is, there exists a perturbation function $\eta : \mathcal{M} \to \mathcal{X}$ with a small norm (the size of the perturbation) such that every point in $\mathcal{X}$ can be written as $x + \eta(x)$ for some $x \in \mathcal{M}$. Then the approximation is valid as long as the scale parameter $s$ remains larger than the size of the perturbation.*

Since we have that the size of our perturbations is bounded by $\epsilon^2 < s$, this gives us the desired result.

Next we show pointwise convergence holds for arbitrary resampling $c$. This is a simple consequence of the linearity of the operator $L$. We order the samples as $x_{i,j}$, where $i$ indexes the original data points $i = 1, ..., N$, and $j$ indexes the resample $j = 1, ..., c$. Furthermore, let $L_c^j$ denote the Laplacian operator defined only on the $j$th resampled points $x_{i,j}$ for $i = 1, ..., N$. Then

$$L_c f(x) = \frac{1}{cN} \sum_{i=1}^{cN} k_s(x, x_i) f(x) - \frac{1}{cN} \sum_{i=1}^{cN} k_s(x, x_i) \tag{17}$$

$$= \frac{1}{c} \sum_{j=1}^{c} \left[ \frac{1}{N} \sum_{i=1}^{N} k_s(x, x_{i,j}) f(x) - \frac{1}{N} \sum_{i=1}^{N} k_s(x, x_{i,j}) f(x_{i,j}) \right] \tag{18}$$

$$= \frac{1}{c} \sum_{j=1}^{c} L_c^j f(x) \tag{19}$$

$$\xrightarrow[n \to \infty]{} Lf(x) \tag{20}$$

where the last line follows from the first half of the proof.

$\square$

**Remarks.** We have shown that $L'$ converges pointwise to the discrete operator $L$. However, for $L$ to converge to its continuous counterpart in Equation 1 (i.e., the Laplace-Beltrami operator), we actually need to take the scale of the kernel $s$ to zero (see, e.g. the proofs of convergence in Belkin and Niyogi (2008); Hein et al. (2005; 2007)). Then Proposition 5.1 implies that we also need to take $\epsilon$ to zero, so that in the limit we are just sampling from the unperturbed manifold. In fact, it is possible to prove convergence without taking $s$ to zero, however the continuous operator that is recovered no longer refers to the gradient but rather the value of the kernel over the manifold (Von Luxburg et al. (2008) analyze this case). The spectra of these operators are often used in various clustering algorithms; in the context of deep neural networks, these operators yield similar regularizers for "smoothness", though for a different definition than the one used in this work. Finally, it should also be noted that the convergence of these operators depends on the intrinsic dimension of the manifold; as we consider here an $\epsilon$-neighborhood of the manifold which has the same dimension as the ambient space, our operators will converge more slowly (Wang (2015)).

*Proof of Proposition 5.2.* We introduce $\bar{L}_\epsilon := L - L_\epsilon$, which is just the Laplacian on the complement of the $\epsilon$ neighborhood subgraph, i.e., the subgraph whose edges are at least $\epsilon$. Clearly $R(L) = R(\bar{L}_\epsilon) + R(L_\epsilon)$. Then we have the following bounds:

$$R(\bar{L}_\epsilon) = \sum_{i,j \notin S} (f(x_i) - f(x_j))^2 L_{i,j} \leq c_2 N^2 \exp(-m^2) \tag{21}$$

$$R(L_\epsilon) = \sum_{i,j \in S} (f(x_i) - f(x_j))^2 L_{i,j} \geq ac_1 N \exp(-1) \tag{22}$$

where the $a$ is a constant introduced for the $O(N)$ edges in the graph of $L_\epsilon$.

Take $b = (ac_1/c_2)(\exp(m^2 - 1)/N)$. A simple rearrangement gives $c_2 = (b^{-1}ac_1)(\exp(m^2 - 1)/N)$, thus we have that

$$R(L) = R(L_\epsilon) + R(\bar{L}_\epsilon) \tag{23}$$

$$\leq R(L_\epsilon) + c_2 N^2 \exp(-m^2) \tag{24}$$

$$\leq R(L_\epsilon) + \left[(b^{-1}ac_1)(\exp(m^2 - 1)/N)\right] N^2 \exp(-m^2) \tag{25}$$

$$= R(L_\epsilon) + b^{-1}ac_1 N \exp(-1) \tag{26}$$

$$\leq R(L_\epsilon) + b^{-1}R(L_\epsilon) \tag{27}$$

$$= (1 + b^{-1})R(L_\epsilon) \tag{28}$$

Then the first inequality follows from (24), the second inequality is trivial, and the third inequality follows from (23)–(28). □

## B EXPERIMENTAL METHODS AND HYPERPARAMETERS

We use a PreActResNet18 model (He et al., 2016) for the CIFAR-10 robustness experiments. We train using SGD with a batch size of 128 and weight decay $\gamma_K$ of 5e-4. We follow Maini et al. (2019) for our learning rate schedule, which is piecewise linear and starts at 0 and goes to 0.1 over first 40 epochs; 0.005 over the next 40 epochs; and 0 over the final 20 epochs. We increase epsilon from 2 to 8 over epochs 10 to 35. We start the weight $\gamma_I$ of the manifold regularization at 0.8 and the weight $\gamma_H$ of the Hamming regularization at 2,400; these increase linearly up to a factor of 10 from epochs 20 to 80. We set the hyperparameter $\alpha = 8$. We use this set of hyperparamaters for all the CIFAR-10 ablation studies (except when setting $\gamma_I$ or $\gamma_H$ to 0 for the ablation studies involving the individual regularizers).

For the dense regularizer, we used heat kernel with scale $s = \epsilon^2$ as weights for the Laplacian. We use a smaller batch size of 40 to offset the increased memory requirements of storing and computing the dense regularizer.

We use a two-layer convolutional neural network for the CIFAR-10 verification experiments, consisting of 2x2 strided convolutions with 16 and 32 filters, then a 128 hidden unit fully connected layer. This is the same model as used in Wong et al. (2018) and Xiao et al. (2019), except those works use a 100 hidden unit fully connected layer. We use the same schedule for the learning rate and $\epsilon$ as in the PreActResNet18 model. The weight $\gamma_I$ starts at 0.4 and the weight $\gamma_H$ starts at 9000; these increase linearly up to a factor of 10 from epochs 20 to 80. We use the same hyperparameter $\alpha$ as for the PreActResNet18 model.

We use the CNN with four convolutional layers plus three fully-connected layers from Carlini and Wagner (2017) for the MNIST robustness experiments. We use the same schedule for the learning rate, $\epsilon$, $\gamma_I$, and $\gamma_H$ as in the PreActResNet18 model (except that $\epsilon$ scales to 0.3). We use the same hyperparameter $\alpha$ as for the PreActResNet18 model.

To set the hyperparameters $\gamma_I$ and $\gamma_H$ on the PreActResNet18 model, we ran a grid search for $\gamma_I = 0.2, ..., 1.0$ and $\gamma_H/\gamma_I = 100, 200, ..., 500$ and selected the settings which yielded the best robust accuracy against a 20-step $\ell_\infty$ PGD adversary with 10 restarts for $\epsilon = 8/255$ on the full training set; the range of $\gamma_H/\gamma_I$ was set such that the corresponding losses were roughly equal for a randomly initialized, untrained network. For reporting results, we train five models using the selected hyperparameters and report results using the one with the median performance on the test set against a 20-step $\ell_\infty$ PGD adversary at $\epsilon = 8/255$ on CIFAR-10 or 0.3 on MNIST. For the PreActResNet18 model, robust accuracy over the 5 runs ranged from 40.1% to 41.5% with median 40.5%; clean accuracy ranged from 66.9% to 72.4% with median 70.0%.

For our stability results, we use the full version of AutoAttack+, an ensemble of attacks proposed by Croce and Hein (2020), for the $\ell_2$ and $\ell_\infty$ perturbations. We choose the attack because it is parameter-free, which reduces the possibility of misconfiguration; empirically it is at least as strong as standard PGD, and has been successful in breaking many proposed defenses. For the Wasserstein adversary, we use an implementation by Wong et al. (2019). For comparing with standard PGD, we use a 20-step PGD adversary with 10 random restarts and a step size of $2.5 \cdot \epsilon/20$ as implemented by Engstrom et al. (2019). For verification, we adopt the setup of Xiao et al. (2019), using the MIP verifier of Tjeng et al. (2017), with solves parallelized over 8 CPU cores and the timeout set to 120 seconds.

# C  ADDITIONAL EXPERIMENTS

We plot robustness curves for both CIFAR and MNIST against a standard 20-step PGD adversary with 10 restarts using both $\ell_2$ and $\ell_\infty$ bounds in Figure 2. These results show that, compared with models trained using standard PGD against $\ell_\infty$ perturbations, our methods perform on par (or substantially better, in the case of the $\ell_2$ adversary on CIFAR-10) in a variety of settings, despite the disadvantage of not using the adversary during training. We also used a stronger AutoAttack+ adversary for the CIFAR-10 results in Table 1, which we plot for reference as "ours (AA+)".

For the CIFAR-10 results, we use a PreActResNet18 model trained with manifold regularization on CIFAR-10 at $\epsilon = 8/255$. The curves for PGD are taken from Madry et al. (2017). Our findings mirror those in Schott et al. (2018), namely, that a model trained using PGD on CIFAR-10 for $\ell_\infty$ performs poorly against $\ell_2$ attacks. We also ran a 500-step $\ell_\infty$ PGD adversary with 20 restarts at $\epsilon = 8/255$ against our model, yielding 40.4% robust accuracy, which is plotted as "ours (PGD+)".

For the MNIST results, we use the CNN architecture from Carlini and Wagner (2017), trained with manifold regularization on MNIST at $\epsilon = 0.3$. The curves for PGD are taken from Madry et al. (2017). The difference in performance is much less dramatic in this case, which we attribute to the lower complexity of the task. We note in particular that both methods experience a sharp drop in performance in the $\ell_\infty$ case for $\epsilon$ larger than the one used during training.

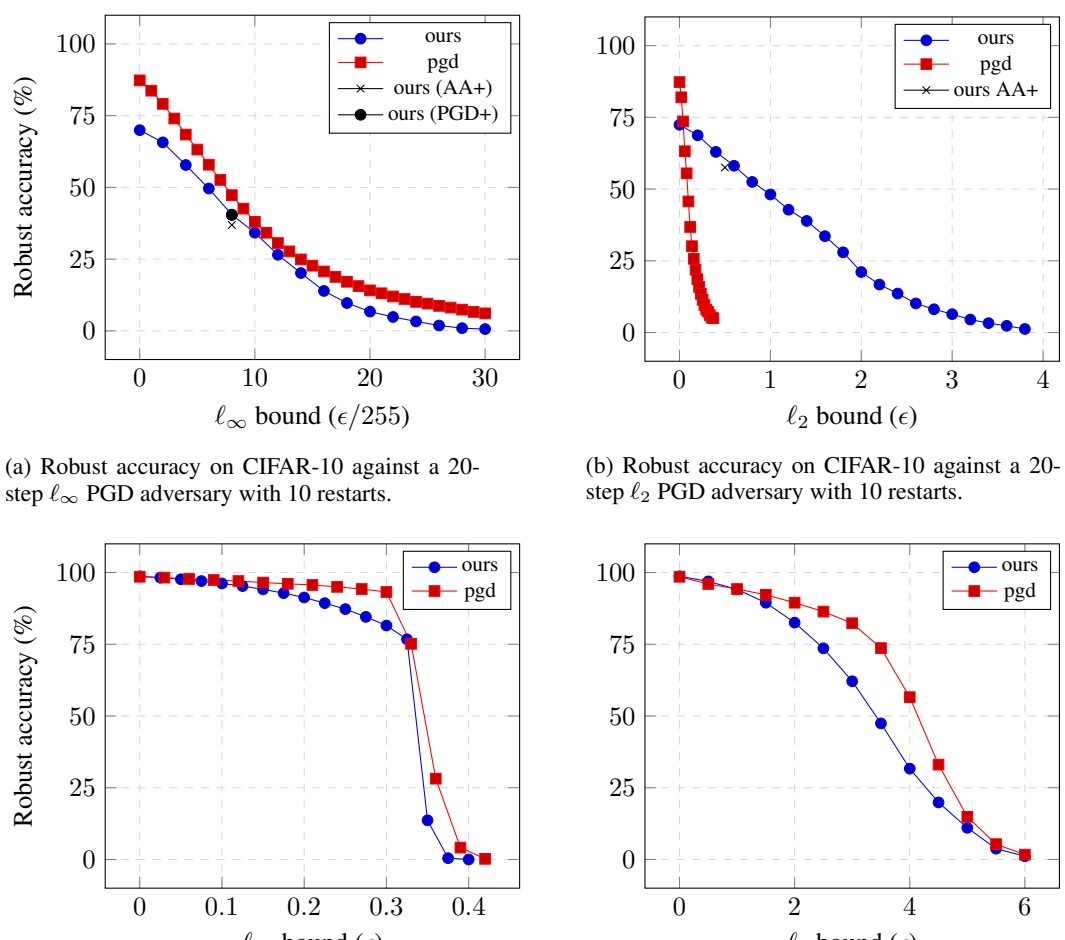

(a) Robust accuracy on CIFAR-10 against a 20-step $\ell_\infty$ PGD adversary with 10 restarts.

(b) Robust accuracy on CIFAR-10 against a 20-step $\ell_2$ PGD adversary with 10 restarts.

(c) Robust accuracy on MNIST against a 20-step $\ell_\infty$ PGD adversary with 10 restarts.

(d) Robust accuracy on MNIST against a 20-step $\ell_2$ PGD adversary with 10 restarts.

Figure 2: Robust accuracy as $\epsilon$ increases for a 20-step PGD adversary with 10 restarts.

