# OpenReview forum: "Manifold Regularization for Locally Stable Deep Neural Networks"
_ICLR.cc/2021/Conference — Reject_

### Official Review · AnonReviewer4 · 2020-10-27
**Interesting ideas but needs better experimental evaluation**

**Rating:** 5
**Confidence:** 4

**Review:**

This work introduces manifold regularization as an approach for learning stable deep nets, towards the goal of adversarial robustness. Several regularizers are proposed: intrinsic, sparse Laplacian and Hamming regularizers. As the proposed method relies only on adding these regularization terms to the loss, it is more computationally efficient than methods that require computation of adversarial examples during training. The proposed method is evaluated on CIFAR-10 under $\ell_2$ and $\ell_{\infty}$ ball attacks and shown to be state-of-the-art in terms of verifiable robustness to $\ell_{\infty}$ attacks at $\epsilon = 8/255$.

Strengths:
- Clear, well-motivated approach
- Hamming regularizer seems to be novel
- State-of-the-art verifiable robustness

Weaknesses:
- Limited experimental evaluation:
  * Lack of comparison with more recent approaches; this could be done by using AutoAttack ("Reliable evaluation of adversarial robustness with an ensemble of diverse parameter-free attacks", ICML 2020)
  * Results from a single epsilon budget and on a single dataset
  * Experiment results seem to be from only a single run
- No direct experimental comparison/validation of stability, which is the stated goal of the work

Overall, this work introduces some interesting and novel ideas, but is lacking in its experimental evaluation, most significantly, in evaluating the stated goal of stability, as opposed to robustness which was done in the experiments. In particular, as noted in the paper, stability can also be achieved by regularizing the Jacobian-norm of the network, which seems like a more direct approach to the goal and should therefore be compared with; indeed, the implementation of the standard manifold regularization term looks almost like a finite difference approximation to the Jacobian-norm.

Other comments:
- The recent work "MACER: Attack-free and Scalable Robust Training via Maximizing Certified Radius", ICLR 2020 also presents an attack-free approach based on randomized smoothing, albeit for achieving certified $\ell_2$ robustness.
- "Disentangling Adversarial Robustness and Generalization", CVPR 2019 suggests that generalization (i.e. on-manifold robustness) is somewhat independent of off-manifold adversarial robustness; the proposed method seems to be aimed at targeting the on-manifold case ("we only need $\epsilon$ stability around valid input points") - what about the off-manifold case?
- Related to the previous point, the sampling procedure in Section 5.4 of picking perturbed points (using "random maximal perturbations") does not seem to restrict samples to be on the data manifold.

*** Post response comments ***

Thanks for the detailed responses and clarifications, especially regarding the manifold used. I suggest that the notion of $\epsilon$-manifold regularization is made clear upfront in the manuscript (e.g. in the introduction) to avoid misunderstanding.

The new results with the dense $\epsilon$-manifold regularizer obtaining 5% less accuracy suggests to me that the specific approximation scheme used is indeed responsible for at least part of the benefit, and it would be useful to understand precisely why this is so since this is the core-contribution of the paper. I also agree with the other reviewers that the overall method as implemented seems a bit disconnected from the core idea of manifold regularization, and rather appears to be a variant of stability training (Zheng et al. 2016). As such, I will be keeping my original rating; nonetheless, I want to again thank the authors for the interesting and vigorous discussion.

---

> ### Author Response · Authors · 2020-11-14
> **1/2**
>
> > Limited experimental evaluation:
>
> Our goal is to train models that are generally robust against multiple (unseen) adversaries. For this goal, the most relevant result is the performance of a single model in multiple adversarial settings, i.e., against state-of-the-art adversaries with standard perturbation bounds (Table 1). We agree that other results may be of interest, and as noted below, Appendix C, Additional Experiments, contains more results addressing a variety of other scenarios.
>
> > Lack of comparison with more recent approaches; this could be done by using AutoAttack ("Reliable evaluation of adversarial robustness with an ensemble of diverse parameter-free attacks", ICML 2020)
>
> We use the full AutoAttack+ adversary with the standard settings for our main results for $\ell_2 = 0.5$ and $\ell_\infty = 8/255$ robustness on CIFAR-10, as noted in Table 1.
>
> > Results from a single epsilon budget and on a single dataset
>
> Appendix C, Additional Experiments, gives comprehensive robustness curves on both CIFAR-10 and MNIST. Our results compare favorably against standard adversarial training using an $\ell_\infty$ PGD adversary, and outperform significantly for $\ell_2$ robustness on CIFAR-10.
>
> > Experiment results seem to be from only a single run
>
> As mentioned in Appendix B, Experimental Methods and Hyperparameters, we report results as the median of 5 runs.
>
> > No direct experimental comparison/validation of stability, which is the stated goal of the work
>
> Definition 4.1 of stability is not a particularly illuminating metric on its own (observe that the constant function is trivially stable but also useless). Rather, one generally asks for both stability and accuracy, i.e., robustness. We introduce stability only for our theoretical development; for evaluation, we consider robustness (of a single model) against multiple (unseen) adversaries, which is a more general notion of robustness than commonly considered in the literature (i.e., robustness against a specific adversary, which is often incorporated into the training procedure).
>
> > In particular, as noted in the paper, stability can also be achieved by regularizing the Jacobian-norm of the network, which seems like a more direct approach to the goal and should therefore be compared with; indeed, the implementation of the standard manifold regularization term looks almost like a finite difference approximation to the Jacobian-norm.
>
> Theoretically speaking, the main difference between our approach (Laplacian regularization) and Jacobian regularization is how the samples used to estimate the regularizer are produced. While regularizing the Jacobian at a point $x$ indeed requires evaluating the model at $x + \delta$ and $x - \delta$, the meaning of $\delta$ is quite different than the $\epsilon$ used in our work. First, to get an accuracy estimate of the Jacobian, one would want to take $\delta$ small, whereas in our setting $\epsilon$ is a fixed quantity referring to the neighborhood over which we would like to be stable. Secondly, our samples are all drawn from the $\epsilon$-neighborhood, whereas one can check that regularizing the Jacobian over the $\epsilon$-neighborhood actually requires evaluating at $x + \epsilon + \delta$ and $x + \epsilon - \delta$, which are no longer guaranteed to fall within the $\epsilon$-neighborhood. Thus Jacobian regularization (1) requires tuning an additional parameter $\delta$, and also (2) results in regularizing on "off-manifold" directions (here, the manifold of interest is the $\epsilon$-neighborhood, rather than the original data manifold). Manifold regularization also permits a nice representer theorem (as far as we know, Jacobian regularization does not).
>
> From a practical standpoint, the reason we do not include Jacobian regularization in our comparisons is because previous works do not measure robustness in the standard settings for robustness. For instance, we believe the current SOTA using Jacobian Regularization is cited in [1], however their evaluation uses only a weak one-step FGSM adversary, and they report ~50\% robust accuracy on CIFAR-10 at $\ell_\infty = 8/255$ and ~70\% robust accuracy on MNIST at $\ell_\infty=0.2$. In comparison, we achieve 37\% robust accuracy CIFAR-10 at $\ell_\infty = 8/255$ using the strong ensemble AutoAttack+, and 91\% robust accuracy on MNIST at $\ell_\infty=0.2$ using standard 20-step PGD adversary with 10 restarts.
>
> ==========
>
> [1] Daniel Jakubovitz and Raja Giryes. Improving dnn robustness to adversarial attacks using jacobian regularization. InProceedings of the European Conference on Computer Vision (ECCV), pages 514–529, 2018

---

> > ### Author Response · Authors · 2020-11-14
> > **2/2**
> >
> > > The recent work "MACER: Attack-free and Scalable Robust Training via Maximizing Certified Radius", ICLR 2020 also presents an attack-free approach based on randomized smoothing, albeit for achieving certified robustness.
> >
> > Thank you for the additional reference (we will be sure to update). Note that, despite the fact that they make similar claims re: efficiency ("[because] MACER does not require adversarial attack during training, it runs much faster to learn a robust model"), they report training for 61 hours to obtain their results on CIFAR-10; in comparison, our results are obtained after training for 3 hours on a single GPU. Additionally, in the $\ell_2$ setting on CIFAR-10 at $\epsilon=1.0$, we achieve about 50\% robust accuracy, compared to 38\% certified accuracy using MACER (though as mentioned these results are not directly comparable). Finally, we note that our model is also additionally robust against $\ell_\infty$ and Wasserstein adversaries.
> >
> > > "Disentangling Adversarial Robustness and Generalization", CVPR 2019 suggests that generalization (i.e. on-manifold robustness) is somewhat independent of off-manifold adversarial robustness; the proposed method seems to be aimed at targeting the on-manifold case ("we only need stability around valid input points") - what about the off-manifold case?
> >
> > Our resampling procedure addresses this issue by expanding the manifold used for regularization to the $\epsilon$-neighborhood of the data manifold. Thus our version of on-manifold stability coincides with standard notions of (norm-bounded) adversarial robustness, since the adversary can only produce points that lie within the $\epsilon$-neighborhood of the data manifold.
> >
> > > Related to the previous point, the sampling procedure in Section 5.4 of picking perturbed points (using "random maximal perturbations") does not seem to restrict samples to be on the data manifold.
> >
> > This is correct, and as noted above, this is the crucial point connecting our manifold regularizers with adversarial robustness (as developed in section 5.1).

---

> > > ### Comment · AnonReviewer4 · 2020-11-19
> > > **Is this really manifold regularization?**
> > >
> > > Thanks for the clarifications that address some of my questions and concerns.
> > >
> > > I guess the fundamental issue I am grappling with here is this - fundamentally the paper is motivated from the manifold regularization perspective, but if one were to strictly follow this, it would result in issues with off-manifold adversarial examples as acknowledged in the reply above. The resampling procedure does address this issue, but as also noted by several other reviewers above (R1, R2) - this procedure essentially reduces to Lipschitz regularization, likely via Jacobian regularization, at locations of high data density. To me, this suggests that it is the specific *approximation procedure* that is giving the robustness benefit and through a (known) Lipschitz/Jacobian regularization mechanism, that is somewhat different from the whole idea of manifold regularization.

---

> > > > ### Author Response · Authors · 2020-11-19
> > > > **The relevant manifold is the $\epsilon$-neighborhood**
> > > >
> > > > We appreciate the response. There are some subtleties here that we hope to address--we believe the cosmetic similarities to a well-known regularizer have obscured the conversation.
> > > >
> > > > > it would result in issues with off-manifold adversarial examples as acknowledged in the reply above.
> > > >
> > > > We believe the reviewer is aware of this, but for clarity, our samples are not off-manifold. We define our new manifold to be the $\epsilon$-neighborhood of the original data manifold ($\mathcal{M_\epsilon}$), from which all our samples are drawn. Thus, while the new samples are indeed "off" the original data manifold, they all fall within the new manifold. **This is a very important distinction.**
> > > >
> > > > > this procedure essentially reduces to Lipschitz regularization, likely via Jacobian regularization, at locations of high data density.
> > > >
> > > > In light of the clarification above, we believe the difference between our approach and existing "Jacobian regularizers" is clear.
> > > >
> > > > First, although the notation is certainly suggestive, we do not think one can rigorously apply Jacobian regularization to our setting. To fix notation, let $\delta$ be the finite difference used to approximate the Jacobian, i.e. $||f(x + \delta) - f(x)||^2_2 / \delta^2$. To produce reliable estimates of the Jacobian, one would need to take $\delta$ small. In contrast, we use a large fixed $\epsilon$. This cannot be explained in the framework of Jacobian regularization--for convergence to the correct operator, we would need to take $\epsilon$ to zero, however this means neglecting robustness in the $\epsilon$-neighborhood. Whereas for our convergence results (Section 5.1), $\epsilon$ is fixed and we maintain smoothness over the $\epsilon$-neighborhood. Thus we believe it is more productive to view "approximating" the Jacobian via (large) finite differences as secretly performing our sparse manifold regularization, and not the other way around.
> > > >
> > > > Note that most works with Jacobian regularization simply compute the partials via backprop. [1]
> > > >
> > > > Second, there is another difference with Jacobian regularization in this setting, namely, the notion of "off-manifold" directions as mentioned earlier. If one wanted to penalize the Jacobian on the $\epsilon$-neighborhood, this would mean penalizing the gradient of the function in directions that leave the $\epsilon$-neighborhood: the Jacobian is defined on the ambient space, and does not "see" the manifold $\mathcal{M_\epsilon}$ in the same way that our regularizer does. Practically speaking, this means using a functional of the form $||f(x + \epsilon + \delta) - f(x)||^2_2 / (\epsilon + \delta)^2$. In contrast to our approach, these samples **are** off-manifold.
> > > >
> > > > For more context, the Jacobian regularizer is addressed in the original work by Belkin as "measure-based regularization" [2]. There, another major theoretical limitation is also noted in that such regularizers lack representer theorems (closed-form solutions).
> > > >
> > > > > To me, this suggests that it is the **specific approximation procedure that is giving the robustness benefit** and through a (known) Lipschitz/Jacobian regularization mechanism, that is somewhat different from the whole idea of manifold regularization.
> > > >
> > > > (emphasis added) We are running experiments using the unapproximated regularizer derived in Section 5.1. We believe a result which indicates similar performance with the sparse approximation would confirm that manifold regularization contributes to robustness, while the sparse approximation is more computationally efficient while not affecting robustness too much.
> > > >
> > > > Taking a step back, we believe the community will benefit from our additional perspective (which have, at the very least, provoked a large amount of productive discussion!). The objections raised both here and in other reviews have, by and large, followed from reasoning along intuitive lines; at the very least, the theoretical claims developed in Section 5 have not been disproven. It should also be noted that (to the best of our knowledge):
> > > >
> > > > - despite being far more common in the literature for training deep nets, no work using "Jacobian regularization" comes anywhere close to our empirical results
> > > > - **no other work that does not use adversarial training achieves similar robustness against (standard) norm-bounded adversaries**.
> > > >
> > > > We believe these further suggest that the techniques of manifold regularization have been overlooked, and would be an important contribution to the community.
> > > >
> > > > [1] Daniel Jakubovitz and Raja Giryes. Improving dnn robustness to adversarial attacks using jacobian regularization. InProceedings of the European Conference on Computer Vision (ECCV), pages 514–529, 2018
> > > >
> > > > [2] Mikhail Belkin, Partha Niyogi, and Vikas Sindhwani. Manifold regularization: A geometric framework for learning from labeled and unlabeled examples.Journal of machine learning research, 7(Nov):2399–2434,2006.

---

### Official Review · AnonReviewer3 · 2020-10-28
**The author proposed a regularization-based robust training method. The results shows improvements over the very baseline, but not as good as the current SoTA.**

**Rating:** 4
**Confidence:** 3

**Review:**

Strength:
1. The author introduced a regularization-based robust training method that can perform well on CIFAR-10.
2. The comparison in the experiment section contains many different aspects of robust training.

Weakness:
1. The perturbation of the $\epsilon$-neighborhood on page 4 is not described. Is it Gaussian? Or Uniform? For the Gaussian perturbation, there's another track of verifiable robustness by Cohen "Certified Adversarial Robustness via Randomized Smoothing" and a following-up method to train the robustness called "MACER: Attack-free and Scalable Robust Training via Maximizing Certified Radius".
2. The biggest concern comes from the perturbation part as well. The author includes a local small perturbation and only considers this local information to compute the $L$ matrix, which used to be computed between all the samples. Since the model only considers the arbitrary fake neighbors, the model will have an un-accurate view of the overall manifold (of equation (2)). Otherwise, this sounds like a free-lunch theory. Can the author make a more clear statement that which is given up and which is beneficial? Also, when considering the data manifold, after perturbation, the manifold is almost surely corrupted. For example, if you consider $S^1\subset R^2$, when you perturb by any arbitrary noise, it will turn out to be a ring with dimension $2$ instead of $1$. So I doubt the local perturbation-based estimation will not recover the actual tangent vector, etc., of the manifold.
3. The benefit of this method is not well addressed. For example, compared with the attack-based robust training, does this method performs a faster training speed? How fast? Will this reduce the number of data that is required? Some more discussion on the benefit as well as the compromising will be appreciated.
4. One of the important matrices in this method is $L$. Is this matrix pre-computed or is this matrix computed in each iteration? Will this cause extra storage space?

Some minor comments:
1. The tables should be self-explainable. For example, highlight the SoTA or give a detailed description of evaluating which number is better.
2. The numbering of equations is confusing. The (3)-(6) can be combined and the most important equation (section 5.4) contains no numbering. I would recommend the author to number those important equations.
3. The equation in section 5.4, if expanded, it would be a norm on $||\cdot||_2$ as well as $||\cdot||_H$, which is also the robustness argument this paper is made. Can the author explained why this is different from just constrain those norm?

---

> ### Author Response · Authors · 2020-11-14
> **1/2**
>
> > The perturbation of the $\epsilon$-neighborhood on page 4 is not described. Is it Gaussian? Or Uniform?
>
> For the developments on page 4, the $\epsilon$-neighborhood can be defined with respect to any norm. Any perturbation is allowed so long as it falls in the $\epsilon$-neighborhood, and we would like to bound the worst possible perturbation. In our evaluation, we consider Wasserstein, $\ell_2$, and $\ell_\infty$ perturbations.
>
> > For the Gaussian perturbation, there's another track of verifiable robustness by Cohen "Certified Adversarial Robustness via Randomized Smoothing" and a following-up method to train the robustness called "MACER: Attack-free and Scalable Robust Training via Maximizing Certified Radius".
>
> Thank you for the additional reference (we will be sure to update). Note that, despite the fact that they make similar claims re: efficiency ("[because] MACER does not require adversarial attack during training, it runs much faster to learn a robust model"), they report training for 61 hours to obtain their results on CIFAR-10; in comparison, our results are obtained after training for 3 hours on a single GPU. Additionally, in the $\ell_2$ setting on CIFAR-10 at $\epsilon=1.0$, we achieve about 50\% robust accuracy, compared to 38\% certified accuracy using MACER (though as mentioned these results are not directly comparable). Finally, we note that our model is also additionally robust against $\ell_\infty$ and Wasserstein adversaries.
>
> > The biggest concern comes from the perturbation part as well.
>
> Please see the general responses 1-3) for a detailed discussion.
>
> > The author includes a local small perturbation and only considers this local information to compute the L matrix, which used to be computed between all the samples.
>
> This is not true when we prove convergence in section 5.1. In Section 5.2, we show that, under certain sparsity assumptions in the input data, there is significantly more information contained in the local perturbations, which allows us to drop the other terms in the Laplacian.
>
> > Since the model only considers the arbitrary fake neighbors, the model will have an un-accurate view of the overall manifold (of equation (2)). Otherwise, this sounds like a free-lunch theory. Can the author make a more clear statement that which is given up and which is beneficial?
>
> Convergence of the resampling procedure is provided in Section 5.1, but this requires infinite samples. In Section 5.2, we assume finite samples, and develop a sparsified Laplacian. Thus, we give up a theoretical notion of convergence in exchange for a much more efficient regularizer that still performs well in practice. Finally, as argued in the general response 3), the assumptions Section 5.2 apply in practical settings.
>
> > Also, when considering the data manifold, after perturbation, the manifold is almost surely corrupted. For example, if you consider S1 in R2, when you perturb by any arbitrary noise, it will turn out to be a ring with dimension 2 instead of 1.
>
> This is a key observation which motivates our development in Section 5.1. Instead of computing the Laplacian over samples from the data manifold, we perturb our samples and compute a Laplacian over the perturbed inputs (which are now drawn, in the given example, from the ring, rather than S1). Proposition 5.1 shows that this resampling procedure yields a regularizer which still converges nicely.
>
> > So I doubt the local perturbation-based estimation will not recover the actual tangent vector, etc., of the manifold.
>
> Identifying the tangent vector is not relevant for our method because we only need points from the (perturbed) manifold (Section 5.1).
>
> > The benefit of this method is not well addressed. For example, compared with the attack-based robust training, does this method performs a faster training speed?
>
> Yes.
>
> > How fast?
>
> We obtain our results after 3 hours of training on a single GPU, whereas standard adversarial training takes about 70 hours. This is because standard adversarial training requires several iterations of backprop to compute each input batch, whereas we only need random sampling.
>
> > Will this reduce the number of data that is required?
>
> The goal of our work is to use the same amount of data and learn a model which is simultaneously robust against multiple adversaries, compared to existing approaches which are only robust against a specific adversary. Our results show that we achieve this goal (Table 1).

---

> > ### Author Response · Authors · 2020-11-14
> > **2/2**
> >
> > > Some more discussion on the benefit as well as the compromising will be appreciated.
> >
> > In addition to faster training times, we also get a single model which is robust against multiple adversaries. However, in exchange for this general robustness, it is possible to get a model which is more robust against a specific adversary (e.g., by training directly against the adversary). For instance, standard adversarial training using an $\ell_\infty$ adversary achieves better robustness against in the $\ell_\infty$ case but is completely broken by an $\ell_2$ adversary (Table 1).
> >
> > > One of the important matrices in this method is L. Is this matrix pre-computed or is this matrix computed in each iteration? Will this cause extra storage space?
> >
> > In theory, L can be precomputed, but requires O(n^2) space to store (n is the number of samples). In Section 5.2, we derive a form that allows us to compute L on the fly in O(n) time (see Equation 7). Later, in Section 5.4, our introduction of the Hamming regularizer allows us to drop L entirely. This is a key contributor to our efficiency.
> >
> > > The tables should be self-explainable. For example, highlight the SoTA or give a detailed description of evaluating which number is better.
> >
> > We will address this issue in the next version by adding a column for aggregate performance against multiple adversaries.
> >
> > > The equation in section 5.4, if expanded, it would be a norm on 2 as well as H, which is also the robustness argument this paper is made. Can the author explained why this is different from just constrain those norm?
> >
> > Our version of the $||\cdot||_H$ norm is expressed via the Euclidean norm as well. Perhaps the reviewer can clarify this question?

---

### Official Review · AnonReviewer2 · 2020-10-28
**promising results but questionable motivation**

**Rating:** 4
**Confidence:** 4

**Review:**

The paper proposes new regularizers for obtaining adversarially-robust models, inspired by the "manifold assumption" that data lies on a low-dimensional manifold. The first regularizer is based a sparsified Laplacian regularizer on the dataset along with random perturbations around each point, while the second is an approximation of the hamming distance between activation maps of a point and its local perturbations.

While the obtained results seem promising, the motivation is questionable, in the sense that the obtained regularizers simply encourage local robustness around each point with random perturbations in all directions, and thus do not really exploit anything about a possible manifold structure in the data. The authors simply point to the "sparsity" of the data in high-dimensions (from what I understand, this is in the sense of points being much further away from each other compared to the local generated perturbations) in order to reduce the laplacian regularizer to only edges between perturbations of the same point, which leads to a natural regularizer that looks like a simple surrogate for penalizing the lipschitz constant. The Hamming regularizer seems more interesting but also seems unrelated to the motivation of manifold assumption.

This makes lean towards rejection, and I encourage the authors to better motivate the proposed regularizers and further compare to previous works in theory and in practice.

more comments:
- introduction: the meaning of "sparse data" should be clarified
- proposition 5.1: I am not convinced that L and Lc converge to the same operator, as the underlying distributions are different, perhaps the authors mean that the limiting operators are close to each other in operator norm? a rigorous proof is desirable
- p.5 "By the curse of dimensionality ...": this should be clarified
- definition of H_alpha: please elaborate on what inputs are fed into this function.
- section 5.5 "the two properties converge": this is not clear
- Table 1: are all models in a given column evaluated using the same attack? why were these particular baselines chosen? I am not sure I understand what makes the comparison to Wong et al. a strength of the current approach.
- section 6.2 / Table 2: please clarify what "verified robustness" means.
- table 2: are all numbers in a given row obtained for a single model with a fixed choice of hyperparameters? which metric are these optimized on?

** update after rebuttal **

Thank you for the detailed clarifications. I still find that the recommended method used in practice, which only encourages robustness to local random perturbations, is too disconnected from the motivation of manifold regularization, and will thus keep my score.

---

> ### Author Response · Authors · 2020-11-14
> **1/2**
>
> > While the obtained results seem promising, the motivation is questionable, in the sense that the obtained regularizers simply encourage local robustness around each point with random perturbations in all directions, and thus do not really exploit anything about a possible manifold structure in the data.
>
> Please see the general responses 1) and 2), which thoroughly address this point. Our regularizers are only evaluated around input points, i.e., on the data manifold, and so by construction exploit the manifold assumption. Note that even standard manifold regularization does not require a global view of the manifold structure to work.
>
> > The authors simply point to the "sparsity" of the data in high-dimensions (from what I understand, this is in the sense of points being much further away from each other compared to the local generated perturbations) in order to reduce the laplacian regularizer to only edges between perturbations of the same point, which leads to a natural regularizer that looks like a simple surrogate for penalizing the lipschitz constant.
>
> This is the correct interpretation of sparsity. Please see the general response 3) for additional experimental evidence which support our claims.
>
> The main difference between our development and Lipschitz regularization is that the Lipschitz constant is defined over the entire input space (i.e., it is an ambient regularizer). Indeed, the entire motivation behind intrinsic regularization is that we would like to avoid over-regularizing on portions of the input space that do not matter. We achieve this by regularizing only on points drawn from the manifold (note that our notion of the manifold here is the $\epsilon$-neighborhood, rather than the original data manifold).
>
> > The Hamming regularizer seems more interesting but also seems unrelated to the motivation of manifold assumption.
>
> There are no technical barriers to applying either intrinsic regularizer as ambient regularizers instead. This would involve evaluating them at points sampled at random from the input space (i.e., pure noise), rather than just the input points. For reasons detailed above, however, we think this is unlikely to yield good results (intuitively, for these regularizers, the only solution which achieves zero loss over the entire ambient space is the constant function, whereas regularizing over the intrinsic space still allows the function to vary off the manifold).
>
> > introduction: the meaning of "sparse data" should be clarified
>
> Thank you. See general response 3) for an empirical demonstration of "sparse data".
>
> > proposition 5.1: I am not convinced that L and Lc converge to the same operator, as the underlying distributions are different, perhaps the authors mean that the limiting operators are close to each other in operator norm? a rigorous proof is desirable
>
> Appendix A, Convergence Results, gives a rigorous proof that L and Lc converge pointwise to the same discrete operator. The key is that the resampling procedure can be treated as a sort of noise, whose effect is bounded by a higher-order term. Note that convergence in operator norm requires uniform convergence, which is stronger than pointwise convergence. We will expand the proof to give more details.
>
> > p.5 "By the curse of dimensionality ...": this should be clarified
>
> Thank you. We will rewrite this section with more precise claims and results.
>
> > definition of H_alpha: please elaborate on what inputs are fed into this function.
>
> The inputs are the ReLU pre-activations.
>
> > section 5.5 "the two properties converge": this is not clear
>
> When $\epsilon$ is zero, all perturbations remain on the data manifold, so standard manifold regularization suffices.

---

> > ### Author Response · Authors · 2020-11-14
> > **2/2**
> >
> > > Table 1: are all models in a given column evaluated using the same attack? why were these particular baselines chosen?
> >
> > In Table 1, each row uses a single trained model, and each column uses the same attack setting (implementations may differ, but in all cases our results use the strongest method). Our intention is to evaluate whether a single model is robust against multiple adversaries.
> >
> > We chose these particular baselines, as to the best of our knowledge, these are the only results in the literature which report robust accuracy against the two most common adversaries at standard settings (i.e., on CIFAR-10 with bounds of $\ell_2 = 0.5$ and $\ell_\infty=8/255$). Though there are a handful of other works which consider multiple adversaries, the evaluation uses weaker adversaries and settings (e.g., [4] considers $\ell_\infty=4/255$ and $\ell_1$ robustness, [5] tests only on MNIST, and [6] uses $\ell_\infty=2/255$ and $\ell_2=0.1$).
> >
> > > I am not sure I understand what makes the comparison to Wong et al. a strength of the current approach.
> >
> > We report Wong to provide a baseline for the Wasserstein adversary, though they do not evaluate for $\ell_\infty$ or $\ell_2$ robustness (we are not aware of any prior works which evaluate both $\ell_\infty$ and Wasserstein robustness). Our method also only takes 3 hours to train, whereas Wong uses an expensive 50-step inner optimization for each training batch (they do not report total training time).
> >
> > > section 6.2 / Table 2: please clarify what "verified robustness" means.
> >
> > An input is verifiably robust if a the neural network (as a program) has been verified to give the correct output in the relevant neighborhood. Verified robustness is the percentage of inputs which are verified to be robust. Following the set up of [2], we use the MILP solver from [3] to verify our network.
> >
> > > table 2: are all numbers in a given row obtained for a single model with a fixed choice of hyperparameters? which metric are these optimized on?
> >
> > In Table 2, all the numbers in a given row are indeed obtained from a single model. Hyperparameters were optimized for verified accuracy when applicable, and robust accuracy otherwise.
> >
> > ==========
> >
> > [2] Kai Xiao, Vincent Tjeng, Nur Muhammad Shafiullah, and Aleksander Madry.  Training for faster adversarial robustness verification via inducing relu stability. ICLR, 2019.
> >
> > [3] Vincent Tjeng,  Kai Xiao,  and Russ Tedrake.   Evaluating robustness of neural networks with mixed integer programming. arXiv preprint arXiv:1711.07356, 2017.
> >
> > [4] Florian Tramèr, Dan Boneh. Adversarial Training and Robustness for Multiple Perturbations. NeurIPS 2019.
> >
> > [5] Schott, L., Rauber, J., Bethge, M., and Brendel, W. Towardsthe first adversarially robust neural network model on MNIST. In International Conference on Learning Representations, 2019. URL https://openreview.net/forum?id=S1EHOsC9tX.
> >
> > [6] Croce,  F.  and  Hein,  M.Provable  robustness  against all   adversarial   lp-perturbations   for   p≥1. CoRR,abs/1905.11213,  2019.   URLhttp://arxiv.org/abs/1905.11213.

---

### Official Review · AnonReviewer1 · 2020-10-28
**Interesting ideas that could be further developed.**

**Rating:** 5
**Confidence:** 4

**Review:**

Summary of the paper:
This work aims at proposing a new type of regularization based on the affinity of the samples in the training set which is used, that allows to design more robust neural networks. 3 ideas are combined to obtain a tractable training procedure. First, the authors assume that the data lay on a manifold and propose to be stable to an $\epsilon$-sausage of the manifold rather than the manifold itself. Then, the authors propose to neglect the correlation terms between different samples and to assume that the affinity matrix is diagonal - they however employ some "ghost" samples for computing the affinity. Finally, simple metrics (like hamming distances) are used to obtain a simple to compute regularization term. The authors demonstrate good performances against standard attack, on CIFAR-10.

Pros:
- Interestingly, some of the assumptions on the manifold look reasonable, and it is interesting to rather study a neural network with the distance induced by a manifold rather than with the Euclidean distance of the ambient space. Indeed, the natural setting of the stability is along the manifold, where any points outside the manifold is an outlier: this leads to a different notion of manifold.
- The performances seem good and to use standard approach, and seems fast (though no benchmark is done in the paper)

Cons:
- Unfortunately, in the current writing, I noticed sometimes a certain lack of rigor, which makes the paper difficult to read and the drawn conclusions difficult to understand.
- In my current understanding of the paper, the data manifold isn't really used and the method remains local, and employs solely the Euclidean metric, which sounds paradoxical given the promises of the paper: if two data points are in the same neighborhood, this information can *not* be used by the current algorithm.

Specific remark:
- I don't understand the path from "adversarial robustness" to the definition and proposition 4.1. Indeed, I thought the definition 4.1 was the standard notion of robustness (as used e.g., in https://storage.googleapis.com/pub-tools-public-publication-data/pdf/45227.pdf ) "We reframe the goal of learning with decouples from accuracy" thus sounds slightly wrong, because I believe this was done in previous works.
- Sec 5.2 needs a substantial rewriting: "by the curse of dim... grows" sounds really bad. I don't understand the equation right above eq. (7), what is L'? This is not introduced. "the curse of dimensionality implies data becomes more sparse" sounds really wrong. On the contrary, sparsity is an assumption to fight the curse of dimensionality. I think the idea can be understood as neglecting the cross term of the covariance matrix, under a local approximation. I find the final justification very informal, and I think there is a confusion between sparsity and high dimension.
- Sec 5.3 and 5.4 describe experimental protocols that include a lot of new hyper parameter. I'd be curious and happy to see the cross validation protocol and its results. For instance, it's unclear to me what is the conceptual difference between the 3 regularizition-distances proposed and why/how, in a joined manner, they affect the performances.
- As described in 5.5, the validity of the model indeed requires on a certain notion of invariance (or smoothness), which is linked  strongly to a manifold assumption, and the idea that, the l2 metric is important, is not new. In fact, it is somehow limited as translation is also a major variability and one could used the quotient distance related to this Lie group over the manifold of images. Furthermore, going beyond this type of explicit & analytic distance(eg l2 metric) is one of the purpose of graph-based regularisation, but according to this paper, this would require to understand the regularity of the data because there is no clear model of the manifold $\mathcal{M}$ of the data.
- The notion of regularity which is used here doesn't introduce more complex smoothness that what is used traditionally in the litterature of adversarial examples for neural network: it's almost purely an additive perturbation that relies on an euclidean distance.

Suggestion for improving this paper:
- I would try to find a manner to employ a non-diagonal approximation of the Laplacian or at least to verify this case is the appropriate one in practice. Indeed, if the data were sampled from the same low-dimensional manifold, then this assumption would be too strong.
- I believe it is important to clarify the experimental protocol and to clean the informal statements.
- I would also try to go beyond the Euclidean setting.

---

> ### Author Response · Authors · 2020-11-14
> **1/3**
>
> We will address first the suggestions for improvements:
>
> > I would try to find a manner to employ a non-diagonal approximation of the Laplacian or at least to verify this case is the appropriate one in practice.
>
> We provide hard evidence that our approximation is appropriate for the setting of our experiments in general response 3) above. We believe this approximation continues to hold for higher-dimensional datasets due to arguments similar to the one found at the end of section 5.2.
>
> >  Indeed, if the data were sampled from the same low-dimensional manifold, then this assumption would be too strong.
>
> The dimensionality of the data manifold alone does not restrict the distance between points drawn from it.
>
> > I believe it is important to clarify the experimental protocol and to clean the informal statements.
>
> A detailed description of the experimental protocol is provided in Appendix B, Experimental Methods and Hyperparameters. We will clarify our statements in our revision.
>
> > I would also try to go beyond the Euclidean setting.
>
> While this is an interesting direction, our theoretical developments in Section 5 are independent of the particular norm. We conduct our experiments in the Euclidean setting so that we may compare our results with the the standard approaches to adversarial robustness.
>
> > The performances seem good and to use standard approach, and seems fast (though no benchmark is done in the paper)
>
> Training the model for CIFAR-10 robustness takes about 3 hours on single GPU using our methods, compared to several days--roughly 70 hours--for standard adversarial training (Section 6.1). The substantial improvements in speed come from the fact that producing a batch for adversarial training involves running many iterations of backprop, whereas our regularizer relies only on random sampling and is essentially free to compute.
>
> > Unfortunately, in the current writing, I noticed sometimes a certain lack of rigor, which makes the paper difficult to read and the drawn conclusions difficult to understand.
>
> We will tighten the exposition in our final draft. We sought to avoid duplicating the work of Belkin [1] (whence our results follow quite easily) however the remarks have been very helpful in identifying portions where details would be useful.
>
> > In my current understanding of the paper, the data manifold isn't really used and the method remains local, and employs solely the Euclidean metric, which sounds paradoxical given the promises of the paper: if two data points are in the same neighborhood, this information can not be used by the current algorithm.
>
> Please see general responses 1-3, which address this point in full. The main observation is that the resampled points are much closer together than the input data, and so contain more information. In 1), we demonstrate that by only regularizing around input points, we indeed use the structure of the data manifold. In 3) we show that, in fact, the data points in CIFAR-10 almost never fall in the same $\epsilon$-neighborhood. Though this may seem paradoxical, we clarify our position further in 2).
>
> As an aside, the metric we place on the manifold is immaterial to this development; convergence holds for any metric.
>
> ==========
>
> [1] Mikhail Belkin, Partha Niyogi, and Vikas Sindhwani.   Manifold regularization:  A geometric framework for learning from labeled and unlabeled examples.Journal of machine learning research, 7(Nov):2399–2434,2006.

---

> > ### Author Response · Authors · 2020-11-14
> > **2/3**
> >
> > > I don't understand the path from "adversarial robustness" to the definition and proposition 4.1. Indeed, I thought the definition 4.1 was the standard notion of robustness (as used e.g., in https://storage.googleapis.com/pub-tools-public-publication-data/pdf/45227.pdf ) "We reframe the goal of learning with decouples from accuracy" thus sounds slightly wrong, because I believe this was done in previous works.
> >
> > Thank you for the reference--note that the paper actually refers explicitly to stability, not robustness, and we definitely do not claim our definition of stability is novel. However the standard optimization objective in adversarial training explicitly minimizes the loss with respect to the true label in the neighborhood of an input [2], whereas stability only asks that points in the neighborhood have the same output. The difference is subtle but necessary for the convergence arguments to go through.
> >
> > > Sec 5.2 needs a substantial rewriting: "by the curse of dim... grows" sounds really bad. I don't understand the equation right above eq. (7), what is L'? This is not introduced.
> >
> > L' refers to the full, un-approximated Laplacian, whereas L is our sparse approximation--we will clarify this in our final draft. eq 7 says that sparsifying the Laplacian proportionally to edge weights (as we propose) yields an unbiased estimator.
> >
> > > "the curse of dimensionality implies data becomes more sparse" sounds really wrong. On the contrary, sparsity is an assumption to fight the curse of dimensionality.
> >
> > We believe our use of the curse of the dimensionality and sparsity is fairly common in the literature, for instance see [1]: "Another manifestation of the curse is that the sampling density is proportional to $N^{1/p}$, where $p$ is the dimension of the input space and $N$ is the sample size... Thus in high dimensions all feasible training samples sparsely populate the input space."
> >
> > However, to avoid confusion we support this observation with the empirical argument presented in 3).
> >
> > >  I think the idea can be understood as neglecting the cross term of the covariance matrix, under a local approximation.
> >
> > To be precise we do not neglect all the "cross terms", only those arising between the original data points; there are still "cross terms" between the resampled points.
> >
> > >  I find the final justification very informal, and I think there is a confusion between sparsity and high dimension.
> >
> > See 3), which shows that the data points in CIFAR-10 almost never fall in the same $\epsilon$-neighborhood.
> >
> > > Sec 5.3 and 5.4 describe experimental protocols that include a lot of new hyper parameter. I'd be curious and happy to see the cross validation protocol and its results.
> >
> > The process for selecting the hyperparameters is given in Appendix B, Experimental Methods and Hyperparameters.
> >
> > > For instance, it's unclear to me what is the conceptual difference between the 3 regularizition-distances proposed and why/how, in a joined manner, they affect the performances.
> >
> > Conceptually, the ambient regularizer is a standard L1 or L2 regularizer that penalizes over the ambient space. The basic intrinsic regularizer penalizes stability with respect to the outputs, while the Hamming regularizer penalizes stability with respect to the number of different local linear components. The individual contributions to performance are in our ablation studies (Table 1).
> >
> > [1] Hastie, Trevor, Robert Tibshirani, and Jerome Friedman. The elements of statistical learning: data mining, inference, and prediction. Springer Science & Business Media, 2009.

---

> > > ### Author Response · Authors · 2020-11-14
> > > **3/3**
> > >
> > > > As described in 5.5, the validity of the model indeed requires on a certain notion of invariance (or smoothness), which is linked strongly to a manifold assumption, and the idea that, the l2 metric is important, is not new. In fact, it is somehow limited as translation is also a major variability and one could used the quotient distance related to this Lie group over the manifold of images.
> > >
> > > Finding a useable metric which is translation invariant is an interesting but open problem, and we agree there are other natural notions of robustness which we do not capture here. However note that the Wasserstein distance is a metric which often described as closer to human perception, for which reason we include it as an adversary in our evaluations. For training we evaluate our regularizers using the $\ell_2$ norm (though there is no theoretical reason we cannot use more exotic norms).
> > >
> > > > Furthermore, going beyond this type of explicit & analytic distance(eg l2 metric) is one of the purpose of graph-based regularisation, but according to this paper, this would require to understand the regularity of the data because there is no clear model of the manifold of the data.
> > >
> > > One does not need an explicit model of the data manifold to learn functions which are smooth over the data manifold [1]. We also don't see any relationship between having a model of the data manifold, which is a topological space that exists independent of any metric, and our usage of the $\ell_2$ metric.
> > >
> > > > The notion of regularity which is used here doesn't introduce more complex smoothness that what is used traditionally in the litterature of adversarial examples for neural network: it's almost purely an additive perturbation that relies on an euclidean distance.
> > >
> > > To the best of our knowledge, norm-bounded adversaries, particularly the $\ell_\infty$ adversary, are among the most well-studied adversarial models in the literature [2].
> > >
> > > ==========
> > >
> > > [1] Mikhail Belkin, Partha Niyogi, and Vikas Sindhwani.   Manifold regularization:  A geometric framework for learning from labeled and unlabeled examples.Journal of machine learning research, 7(Nov):2399–2434,2006.
> > >
> > > [2] Aleksander Madry, Aleksandar Makelov, Ludwig Schmidt, Dimitris Tsipras, and Adrian Vladu. Towards deep learning models resistant to adversarial attacks, 2017.

---

> > > > ### Comment · AnonReviewer1 · 2020-11-17
> > > > **A short answer**
> > > >
> > > > Dear authors,
> > > > I've read your rebuttal and I thank you for clarifying several points. I'm going to try to keep it short because our thread will be hard to follow.
> > > >
> > > > - CIFAR-10 is a high-dimensional dataset, thus it makes sens that $\mathbb{R}^{3\times 32^2}$ is mainly empty: this is just the curse of dimensionality. In my community, sparsity refers to a low dimensional structure that allows the data to be sparsely and tractably expressed in some basis. Being curse by the dimensionality means on the contrary that a problem is hard. Also, there are some works that show that nearest neighbor(or linear SVM) performance on CIFAR isn't that bad, which is surprising given several statements in the manuscript. I agree up to a certain extend that the issue is a matter of vocabulary.
> > > > - In the sec5.2, the formula between $L'$ and $L$ almost explicitly imply that $L=L'$ (it depends to which subspace $x$ belongs to), thus I'm not sure this is what is really expressed there. Furthermore, the notation $L(x_{i,j},x_{i,k})$ is never introduced, to my knowledge. I still think this statement is not correct, or at least has to be clarified: "By the curse of dimensionality, the approximation becomes increasingly accurate as the number of dimensions grows". I don't think those are "details".
> > > > - "The dimensionality of the data manifold alone does not restrict the distance between points drawn from it.", of course, but the dimensionality of the intrinsic manifold guides the number of neighbors.
> > > > - Locality: I think that locality (ie, few points in a neighborhood) on a manifold is almost the only property that one can use on a manifold, and I think all the reviewers understand this. However, in the proposed method, one relies systematically on a **single** point (which is then resampled). I believe it is this part that is questionable, because this is not specific to manifolds(one could weaker this assumption to a model where the data are a union of $\epsilon$-ball) and it reads like a surrogate of a Lipschitz constant minimization.(R2)
> > > > - "The individual contributions to performance are in our ablation studies (Table 1)." It's slightly unclear to me that each effect can be studied independently.
> > > > - Appendix B: I see only in the 4th paragraph the mention of a validation set (which is the full training set), and otherwise the description looks like the parameters have been adapted to the task without any specific precision. I think this methodology is not right but maybe I misunderstand?
> > > > - I think the connexion with (Belkin et al) should be extremely explicit in the text.
> > > > - I'd be happy to read a corrected version of the paper.

---

> > > > > ### Author Response · Authors · 2020-11-19
> > > > > **Small clarifications**
> > > > >
> > > > > We will be uploading an updated version of the paper shortly, so we will only address a few small concerns here.
> > > > >
> > > > > > In the sec5.2, the formula between $L$ and $L'$ almost explicitly imply that $L = L'$ (it depends to which subspace belongs to), thus I'm not sure this is what is really expressed there.
> > > > >
> > > > > As noted in the previous response (2/3), $L'$ refers to the full, un-approximated Laplacian, whereas $L$ is our sparse approximation. We can imagine taking the underlying graph, and sampling edges according to their weights. Then this is an unbiased estimator of the original Laplacian in the spectral sense (i.e., as a symmetric bilinear operator, as Eq 7 shows).
> > > > >
> > > > > > Furthermore, the notation $L(x_{i,j},x{i,k})$ is never introduced, to my knowledge.
> > > > >
> > > > > $L$ is a matrix generated by a (heat) kernel, so in $L(x_{i,j},x_{i,k})$ we identify the matrix with its kernel (to avoid confusions with the index notation).
> > > > >
> > > > > > I still think this statement is not correct, or at least has to be clarified: "By the curse of dimensionality, the approximation becomes increasingly accurate as the number of dimensions grows". I don't think those are "details".
> > > > >
> > > > > We will clarify this in our updated paper as a rigorous statement which bounds the approximation using various quantities (including a separation between points, $\epsilon$, and the scale parameter of the kernel $s$). The version of the curse we refer to has the distance between points grow as the ambient dimension grows. For a fixed $\epsilon$ we become increasingly likely to sample the edge (due to the exponential form of the kernel), leading to a more precise approximation.
> > > > >
> > > > > > "The dimensionality of the data manifold alone does not restrict the distance between points drawn from it.", of course, but the dimensionality of the intrinsic manifold guides the number of neighbors.
> > > > >
> > > > > We're not sure we follow the implications of this statement.  Note that the the $\epsilon$ neighborhood has full dimension, while the original points are drawn from a manifold of much lower dimension (by assumption).
> > > > >
> > > > > > Locality: I think that locality (ie, few points in a neighborhood) on a manifold is almost the only property that one can use on a manifold, and I think all the reviewers understand this. However, in the proposed method, one relies systematically on a single point (which is then resampled). I believe it is this part that is questionable, because this is not specific to manifolds(one could weaker this assumption to a model where the data are a union of $\epsilon$-ball) and it reads like a surrogate of a Lipschitz constant minimization.(R2)
> > > > >
> > > > > The union of $\epsilon$ balls is indeed a manifold, albeit not connected. Note that the Lipschitz constant is defined on the ambient space, i.e., it controls the behavior of the function everywhere, not just where points are drawn (whether the data manifold is connected or not). In contrast, we only use samples from the ($\epsilon$-neighborhood$) of the manifold. Please let us know if the distinction between ambient and intrinsic regularizers remains unclear.
> > > > >
> > > > > We emphasize that sparsifying Laplacians for use in applications is very standard (see below), so perhaps the reviewer could suggest where exactly in our procedure the reviewer takes issue? Is it our specific sparsification method, our resampling procedure, or the application of manifold regularization in general?
> > > > >
> > > > > We provide the following observation which may help sway the reviewer's intuition. Spectral clustering algorithms often proceed by defining similar measures on the edges of a graph, where the nodes are the data points. One method is to use a heat kernel, which gives our Laplacian. (One could also use a kNN graph, or an $\epsilon$-neighborhood graph by connecting all points that are within $\epsilon$ of each other. See Section 2.2 [1].). Then the top $k$ eigenvalues of the similarity matrix give a way to generate $k$ clusters. In our idealized setting, we naturally have $n$ dense clusters consisting of the resampled points. This suggests that most of the information in the spectrum of the full Laplacian is captured by our sparse approximation. We emphasize that this is only an intuitive illustration; we are working hard to get an updated version of the paper so that hopefully the conversation can focus on more rigorous statements.
> > > > >
> > > > > [1] https://www.cs.cmu.edu/~aarti/Class/10701/readings/Luxburg06_TR.pdf

---

> > > > > > ### Comment · AnonReviewer1 · 2020-11-19
> > > > > > **-**
> > > > > >
> > > > > > "As noted in the previous response (2/3),  refers to the full, un-approximated Laplacian, whereas  is our sparse approximation. "
> > > > > > Can you expand on that equality: $\sum x_iL_{ij}x_j=\mathbb{E}\big[\sum x_iL_{ij}x_j\big]$?
> > > > > >
> > > > > > "We're not sure .."
> > > > > > Note that one of the point raised by R1, R2, R3, R4 concerns the number of neighbor used by the proposed method and the fact that this approximation doesn't rely on an underlying manifold assumption but a weaker assumption.
> > > > > >
> > > > > > "The union of  $\epsilon$-balls is indeed a manifold, albeit not connected."
> > > > > > Note that this manifold has the same dimension as the ambient space and there is thus an explicit curse of dimensionality: the approximate regularizer would converge slowly to $\Vert f\Vert_I$.
> > > > > >
> > > > > > "Note that the Lipschitz constant is defined on the ambient space, i.e., it controls the behavior of the function everywhere, not just where points are drawn"
> > > > > > Note that a Lipschitz constant can be local.
> > > > > >
> > > > > > "Please let us know if the distinction between ambient and intrinsic regularizers remains unclear."
> > > > > > Thanks, I think I know the distinction.
> > > > > >
> > > > > > "the reviewer could suggest where exactly in our procedure the reviewer takes issue"
> > > > > > I think the issue was precisely raised by each reviewer of this paper.

---

> > > > > > > ### Author Response · Authors · 2020-11-19
> > > > > > > **Response**
> > > > > > >
> > > > > > > We would like to sincerely thank the reviewer for engaging in this back-and-forth discussion. We believe such interactions have made our revision much stronger.
> > > > > > >
> > > > > > > > "As noted in the previous response (2/3), refers to the full, un-approximated Laplacian, whereas is our sparse approximation. " Can you expand on that equality: $\sum x_i L_{ij} x_j = \mathbb{E}\bigg[\sum x_i L_{ij} x_j\bigg]$
> > > > > > >
> > > > > > > The notation here is indeed unclear, as we are using $L_{ij}$ to refer to the realized values of the resampled Laplacian on the left, and the random variables in the sampling procedure on the right--we will update this in our revision. For clarity, we are just observing that sampling edges proportionally to their weights gives an unbiased estimator (this follows directly since the Laplacian is just a matrix of edge weights). As this is not actually what we actually propose, we are replacing this with a precise claim about the quality of our approximation under certain sparsity assumptions. The dependence on the dimension of the ambient space will also be more clear... thank you for your patience on this matter!
> > > > > > >
> > > > > > > > "We're not sure .." Note that one of the point raised by R1, R2, R3, R4 concerns the number of neighbor used by the proposed method and the fact that this approximation doesn't rely on an underlying manifold assumption but a weaker assumption.
> > > > > > >
> > > > > > > We are currently running experiments using the full graph, hopefully this can help settle this point.
> > > > > > >
> > > > > > > > "The union of $\epsilon$-balls is indeed a manifold, albeit not connected." Note that this manifold has the same dimension as the ambient space and there is thus an explicit curse of dimensionality: the approximate regularizer would converge slowly to
> > > > > > >
> > > > > > > This is a good observation, and we shall be sure to mention it in our revision. Note that this is not introduced by our resampling procedure but rather that fact that $\epsilon$-stability requires controlling the behavior of the function on the full ambient dimension.
> > > > > > >
> > > > > > > > "Note that the Lipschitz constant is defined on the ambient space, i.e., it controls the behavior of the function everywhere, not just where points are drawn" Note that a Lipschitz constant can be local.
> > > > > > >
> > > > > > > Thank you for the clarification. We would argue that such a local lipschitz constant, depending on how it is evaluated, deserves an analysis using the framework of manifold regularization, which may yield additional insights on its limiting behavior.
> > > > > > >
> > > > > > > > "the reviewer could suggest where exactly in our procedure the reviewer takes issue" I think the issue was precisely raised by each reviewer of this paper.
> > > > > > >
> > > > > > > So far, many (though certainly not all!) of the more persistent concerns have been of the form, "you do not use enough information about the manifold". Our argument is that for a dataset such as CIFAR-10, the information we are given is negligible compared to what is introduced by the resampling procedure. The specific approximation is proposed on heuristic grounds, but follows standard methods in the literature. So we are just hoping to get some specifics as to where exactly the reviewers believe our method discards the relevant information. However, this discussion may be more productive post-revision.

---

### Author Response · Authors · 2020-11-14
**1) local methods still exploit manifold structure**

We thank all the reviewers for their detailed comments and suggestions.

We wanted to use this general response to address in detail some common points raised by the reviewers. In particular, several reviewers had some concerns about local form of the regularizer we ultimately use for training in Equation 7 (R1: "the data manifold isn't really used and the method remains local", R2: "the obtained regularizers simply encourage local robustness around each point with random perturbations in all directions, and thus do not really exploit anything about a possible manifold structure in the data", R3: "The author includes a local small perturbation and only considers this local information", etc.).

The data manifold features prominently in our method, even in the sparse case. First, manifold regularization is inherently a local method, as can be seen when we use exponential weights in the Laplacian. Second, that our regularizers are only evaluated on input data already suggests that the form of the regularizer depends not on the ambient space (e.g., L2 regularization, which penalizes the behavior of the classifier over the entire input space) but rather the intrinsic space (i.e., where the data is drawn). As a nice theoretical bonus we indeed have convergence in the limit to the Laplacian operator over the entire data manifold, however, one should not let this obscure the fact that we are using local information about the manifold even before converging.

---

> ### Author Response · Authors · 2020-11-14
> **2) convergence (Section 5.1) and practical application (Section 5.2) are independent developments**
>
> More generally, please note that the follow sections address fundamentally different issues:
>
> - Section 5.1 shows that one can use a resampling approach to extend manifold regularization on the data manifold to smoothness over the epsilon neighborhood (as required for robustness), with a convergence result in the limit
> - Section 5.2 develops a sparse approximation for settings suitable for practical application to current datasets
>
> In Section 5.1, where we state convergence of the resampled (dense) Laplacian, we do not neglect any of the longer-range dependencies. This is a nice result that shows our regularizers are well-behaved in a certain sense, but the convergence ultimately relies on the infinite sample limit.
>
> Section 5.2 moves us from the theoretical considerations of Section 5.1 to a more practical setting, first dropping the infinite sample assumption. We believe it is fairly standard to apply techniques whose behavior is understood in the limit to finite samples; in fact, as a community we often apply techniques whose behavior is not even fully understood in the limit (e.g., to the best of our knowledge, only recently have there been some partial convergence results for standard adversarial training [1] or even deep nets [2]). In any case, whether such methods survive outside the theory ultimately comes down to the experimental results. Now, we could have stopped at the dense regularizer developed in Section 5.1 (and it seems likely these concerns would not have been raised); however, in Section 5.2 we take the (perhaps unusual) step of more carefully characterizing the behavior of our regularizers given finite samples. In particular, we argue that in the deep learning regime, one can simplify the form of the Laplacian even further, taking a sparse approximation which is far more computationally efficient, while still capturing most of the information of the original dense Laplacian. We emphasize that taking sparse approximations of the Laplacian is very standard practice for improving efficiency (e.g., using the k-nearest neighbors, or sampling proportionally to edge weights)--our main argument is simply that, in this particular setting, where we introduce a well-behaved resampling procedure, assuming some notion of sparsity in the data, there happens to a computationally cheap way to achieve such sparsification.
>
> ==========
>
> [1] Ruiqi Gao, Tianle Cai, Haochuan Li, Liwei Wang, Cho-Jui Hsieh, Jason D. Lee. Convergence of Adversarial Training in Overparametrized Neural Networks. https://arxiv.org/abs/1906.07916.
>
> [2] Arthur Jacot, Franck Gabriel, Clément Hongler. Neural Tangent Kernel: Convergence and Generalization in Neural Networks. https://arxiv.org/abs/1806.07572.

---

> > ### Author Response · Authors · 2020-11-14
> > **3) CIFAR-10 is sparse**
> >
> > Whether this particular approximation (i.e., retaining only the intra-sample edges) holds is ultimately up to the data (in particular, we cannot do this if the data are too dense relative to the perturbations). The end of Section 5.2 seeks to justify this sparsification using CIFAR-10 as an example, with some back-of-the-envelope calculations. However, given the interest, we ran some additional experiments which we believe help settle this point with some hard evidence. From CIFAR-10, we select a random subset of 10,000 (20\%) training samples. For each of these points, we find the closest point within the remaining 49,999 training samples, and compute the distance. Using the $\ell_2$ metric, we found a mean distance of 9.597, a median distance of 9.405, and the distance at the 10th percentile to be 5.867. Conversely, we resample points in an $\ell_\infty$ ball of radius $\epsilon=8/255$, which is contained in an $\ell_2$ ball of radius $\sqrt{3 \cdot 32 \cdot 32} \cdot 8/255 = 1.739$. In fact, only 21 training samples, or 0.21\% of the random subset, have a partner closer than twice the perturbation bounds! In concrete terms, even if one were to compute the full Laplacian, the contributions from the between-sample edges are almost all negligible; in the end, we are happy to give up a distant notion of convergence in exchange for a vastly more efficient regularizer.

---

> ### Comment · AnonReviewer3 · 2020-11-18
> **Local vs. Global**
>
> I would like to thank the authors' effort to clarify those concerns.
>
> However, for me, the logic is slightly chaotic.
>
> Initially, the paper used "data manifold" and used Laplacian (Equation (2)) to discretely approximate the data samples. This method will look more general (or can be said as a larger local view field) of the data manifold. For example, cat_1 in the dataset can be influenced by cat_2, cat_3 ... around it. Those are **different** samples.
> Then, in Sec. 5.2, due to the perturbation is very small, the sparse approximation of the Laplacian will only consider the **same** sample with different perturbation. For example, cat_1+$\varepsilon_1$, cat_1+$\varepsilon_2$, ... cat_1+$\varepsilon_3$. There is nothing to do with other data samples and the data manifold structure is not considered anymore.
>
> So why is the manifold included? Is it just the intuition and then everything just depends on itself without looking at the data manifold?

---

> > ### Author Response · Authors · 2020-11-18
> > **Manifold assumption**
> >
> > We highly appreciate the continued conversation. We are in the process of running some additional ablation studies (described below) which we believe will lend experimental evidence supporting our proposed methodology. We are also working on rewriting several sections of the paper to include more precise statements of our claims, which will be uploaded before the end of the discussion period.
> >
> > > There is nothing to do with other data samples and the data manifold structure is not considered anymore.
> >
> > Though this may just be a difference in terminology, we think it is important clarify that our regularizer strongly depends on the manifold structure. For instance, we can consider an alternative regularization scheme where one simply samples random pairs of points from the input space, and asks that the learned function be smooth everywhere. This would be some version of a Lipschitz loss. We emphasize that we only evaluate our regularizer at points drawn from the $\epsilon$-neighborhood of the data manifold, which is a very strong dependence on the (local) structure of the data manifold.
> >
> > > This method will look more general (or can be said as a larger local view field) of the data manifold.
> >
> > In fact, the infinitesimal structure is *all* that we care about. The relevant notion of distance on a manifold is the geodesic distance, i.e., the distance as seen from the perspective of the manifold (e.g., great circle distance for a sphere, see chapter 3, 9 in [1] for an overview of geodesics). That the discrete Laplacian converges follows partly from the fact that the two metrics are very similar in sufficiently small neighborhoods, but in general they can diverge arbitrarily. This is why standard manifold regularization requires something like Gaussian weights on the Laplacian, which decay exponentially in the distance between points: the longer-range interactions do not carry reliable information about the manifold, and thus should be attenuated.
> >
> > > So why is the manifold included? Is it just the intuition and then everything just depends on itself without looking at the data manifold?
> >
> > We include our analysis to show that the limiting behavior of our resampling procedure is well-behaved, and in fact the regularizer converges to a nice, interpretable form, i.e. the integral of the Laplacian over the manifold. There are additional benefits that we inherit from this framework, the most significant of which is actually a representer theorem for certain kernel-based methods (i.e., a *closed form solution to the optimization problem* in a RKHS) [2]. These results give us confidence that the regularizer behaves as we would expect; without them, we cannot say anything about a function that achieves the minimum (or whether it even converges at all).
> >
> > Finally, as noted at the end of Section 5.2, our approximation ultimately depends on both the structure of the data, as well as $\epsilon$, both of which are given quantities. It is entirely possible that, for certain datasets and $\epsilon$, the sparsified Laplacian discards an non-insignificant amount of information. Because computing and storing the dense Laplacian $L$ is very expensive, in these cases we would suggest employing the standard k-NN approach to sparsifying the Laplacian. However, doing so introduces an additional overhead, and so **in the experiments, we explore the hypothesis that, in certain cases, there is a sufficiently good approximation that is basically free to compute.**
> >
> > We believe these conclusions are supported by our experiments on CIFAR-10 (and MNIST), namely:
> >
> > - CIFAR-10 is sufficiently sparse, which provides justification that our approximation is, numerically speaking, sound
> > - the sparse approximation, which was proposed for purely for efficiency, is 30x faster than either MACER or standard adversarial training
> > - the regularization scheme is successful, as measured by our model achieving good robustness against a variety of unseen adversaries.
> >
> > We are currently running additional experiments using the dense version of the resampled Laplacian as derived in Section 5.1; the results will give performance of the standard regularizer without the sparsification scheme, and show whether the sparsified regularizer is discarding too much information. We will report our results as soon as they are available (note that a single run takes around 24 hours).
> >
> > [1] Carmo, Manfredo Perdigao do. Riemannian geometry. Birkhäuser, 1992.
> >
> > [2] Mikhail Belkin, Partha Niyogi, and Vikas Sindhwani. Manifold regularization: A geometric framework for learning from labeled and unlabeled examples.Journal of machine learning research, 7(Nov):2399–2434,2006.

---

### Author Response · Authors · 2020-11-21
**Summary of updates**

Thanks again to all for the insightful discussion. The conversations has been very useful in identifying portions of our development and experimental results which needed additional elaboration. We have just uploaded a version which includes revisions suggested by the reviewers; the main points are summarized below.

- We have clarified our usage of "sparse" data in the introduction
- We have updated the Appendix A with a detailed proof of Proposition 5.1.
- We have substantially rewritten Section 5.2, where we develop the sparse approximation
    - We introduce a bound on the approximation error of a sparse regularizer in terms of a precise notion of separation in the data samples. This makes it much more clear how the quality of our approximation depends on various asymptotic behaviors.
    - We replace the back-of-the-envelope calculations regarding CIFAR-10 sparsity with an experiment verifying that the separation requirement is met.
- We include an additional ablation study using the dense regularizer. The results show that the dense regularizer is slightly worse than the sparse regularizer in both robust and clean accuracy, though it still gives 35% robust accuracy on CIFAR-10 against $\ell_\infty$ perturbations at $\epsilon=8/255$, which is significantly better than alternative regularization-only approaches in the literature (i.e., 25% in [1]). Training with the dense regularizer also takes about 21.5 hours, compared to 3 hours using the sparse regularizer.

We understand this update comes very late in the review process, so we would deeply appreciate if the reviewers could take the time to read the updated sections (and provide any additional comments). We will try our best to respond to any requests for clarification.

[1] Tianyu Pang, Kun Xu, Yinpeng Dong, Chao Du, Ning Chen, and Jun Zhu.  Rethinking softmax cross-entropy loss for adversarial robustness, ICLR 2020.

---

### Decision · Program_Chairs · 2021-01-07
**Final Decision**

**Decision:**

Reject

**Comment:**

The paper received borderline and negative reviews but has raised many questions and discussions, showing that the paper has some merit. Many concerns were however raised on various aspects of the paper such as mathematical rigor, clarity, and motivation of manifold regularization that is too disconnected from the robustness to local random perturbation which is encouraged by the method. The rebuttal addresses some of these comments and the reviewers have appreciated the detailed answer. Yet, it was not sufficient to change the reviewer's opinions.

In its current form, the paper is not ready for publication and the area chair agrees with most of the reviewer's comments. He recommends a reject, but encourage the authors to take into account the feedback from the reviewer before resubmitting to a future venue.